# Learn Singularly Perturbed Solutions via Homotopy Dynamics

**Chuqi Chen**[1]   **Yahong Yang**[2]   **Yang Xiang**[1 3]   **Wenrui Hao**[2]

## Abstract

Solving partial differential equations (PDEs) using neural networks has become a central focus in scientific machine learning. Training neural networks for singularly perturbed problems is particularly challenging due to certain parameters in the PDEs that introduce near-singularities in the loss function. In this study, we overcome this challenge by introducing a novel method based on homotopy dynamics to effectively manipulate these parameters. From a theoretical perspective, we analyze the effects of these parameters on training difficulty in these singularly perturbed problems and establish the convergence of the proposed homotopy dynamics method. Experimentally, we demonstrate that our approach significantly accelerates convergence and improves the accuracy of these singularly perturbed problems. These findings present an efficient optimization strategy leveraging homotopy dynamics, offering a robust framework to extend the applicability of neural networks for solving singularly perturbed differential equations.

## 1. Introduction

The study of Partial Differential Equations (PDEs) serves as a cornerstone for numerous scientific and engineering disciplines. In recent years, leveraging neural network architectures to solve PDEs has gained significant attention, particularly in handling complex domains and incorporating empirical data. Theoretically, neural networks have the potential to overcome the curse of dimensionality when solving PDEs (Han et al., 2018; Siegel & Xu, 2020; Lu et al., 2021b;

Yang & Xiang, 2022; Hao et al., 2025). However, despite these advancements, numerically solving such fundamental physical equations remains a challenging task. Existing neural network-based PDE solvers can be broadly divided into two categories. The first is solution approximation, which focuses on directly approximating PDE solutions using methods such as PINNs (Raissi et al., 2019; Karniadakis et al., 2021; Cuomo et al., 2022), the Deep Ritz Method (E & Yu, 2018), and random feature models (Chen et al., 2022; Dong & Wang, 2023; Sun et al., 2024; Chen et al., 2024b). The second category, operator learning, aims to approximate the input-to-solution mapping, with representative methods including DeepONet (Lu et al., 2021a) and FNO (Li et al., 2021), as well as various extensions for broader operator classes (He et al., 2024; Lan et al., 2023; Li et al., 2023; Geng et al., 2024).

However, the optimization challenges in solving PDEs significantly limit the applicability and development of neural network-based methods. Studies have shown that the loss functions for solving PDEs are often difficult to minimize, even in simple scenarios (Krishnapriyan et al., 2021; Rathore et al., 2024; Xu et al., 2024; Chen et al., 2024b;a). For example, small diffusion coefficients in the Allen–Cahn equation (Allen & Cahn, 1975), small viscosity terms in the Burgers equation (Burgers, 1948), and large wave numbers in the Helmholtz equation (Hilbert, 1985). In these types of equations, the parameters significantly influence the solution behavior. In Allen–Cahn and Burgers equations, decreasing the parameter sharpens the solution, often resulting in near-singular structures. In the Helmholtz equation, increasing the parameter induces high-frequency oscillations. These effects complicate the loss landscape, making optimization challenging and often causing slow convergence, inaccurate solutions, or even divergence.

The root of this challenge lies in the highly complex energy landscape of the loss function near singularities, which significantly exacerbates optimization difficulties (Karniadakis et al., 2021; Xu et al., 2024). To address these challenges, two main strategies have been proposed. The first strategy is resampling, which involves introducing additional collocation points in regions with low regularity to better capture the solution's complexity (Wight & Zhao, 2020; Gao et al., 2024; Zhang et al., 2025). However, resampling-based methods typically require a large number of sample

---

[1]Department of Mathematics, The Hong Kong University of Science and Technology, Clear Water Bay, Hong Kong SAR, China [2]Department of Mathematics, The Pennsylvania State University, PA, USA [3]Algorithms of Machine Learning and Autonomous Driving Research Lab, HKUST Shenzhen-Hong Kong Collaborative Innovation Research Institute, Futian, Shenzhen, China . Correspondence to: Yahong Yang <yxy5498@psu.edu>.

*Proceedings of the 42nd International Conference on Machine Learning*, Vancouver, Canada. PMLR 267, 2025. Copyright 2025 by the author(s).

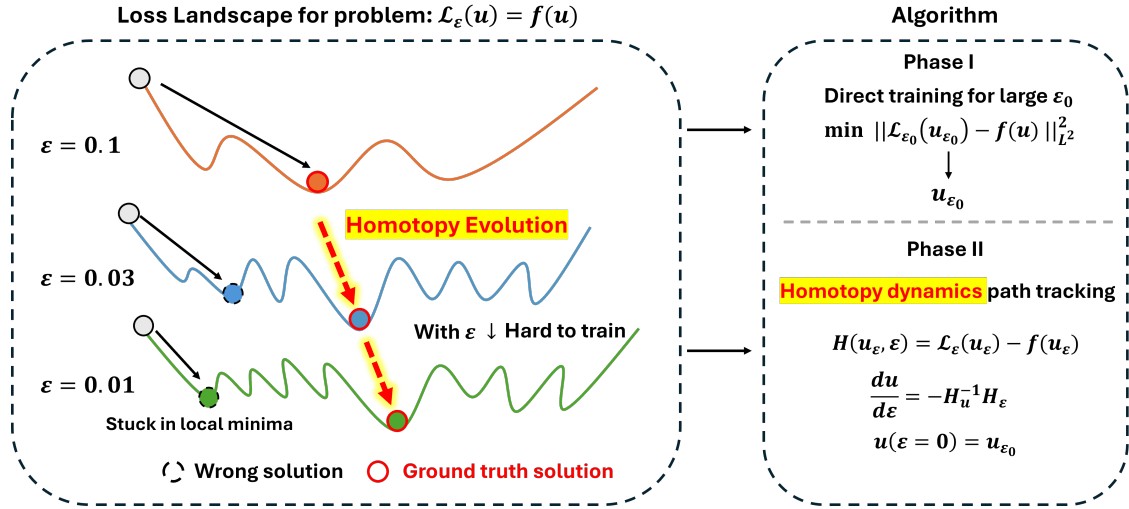

*Figure 1.* Framework of homotopy dynamics for solving singularly perturbed problems.

points, leading to substantial memory consumption, and the sampling process becomes increasingly complicated in high-dimensional settings. The second strategy is the design of multiscale neural network architectures (Wang, 2020; Liu, 2020; Liu et al., 2024b; Hao et al., 2024; Wang et al., 2021; Huang et al., 2025). These approaches generally require certain a priori knowledge of the solution properties, impose specific constraints on the network design, and are highly sensitive to the selection of hyperparameters.

In this paper, we introduce a novel approach based on homotopy dynamics to gradually reshape the complex energy landscape with respect to a specific coefficient. Rather than directly computing solutions near singularities, we leverage homotopy dynamics to trace a solution path that approximates them more effectively. More specifically, we investigate the training challenges introduced by a parameter $\varepsilon$ in the PDE residual term within the loss functions. As $\varepsilon$ decreases, the problem becomes more significantly difficult to solve. To understand this effect, we provide a theoretical analysis of how $\varepsilon$ influences the convergence of the training process. To address this issue, we propose a novel method called *Homotopy Dynamics*. The key idea is to first train the neural network on PDEs with a large $\varepsilon$, where the problem is easier to learn and training is more efficient. Then, we gradually and adaptively adjust the neural network according to the evolution of the homotopy dynamics, guiding $\varepsilon$ toward its target value (as illustrated in Figure 1). Although the homotopy approach has been used to train neural networks (Chen & Hao, 2019; Yang et al., 2025), this work is the first to apply homotopy dynamics to sharp interface problems in PDEs through the parameter $\varepsilon$.

A related idea appears in (Krishnapriyan et al., 2021) as Curriculum PINN Regularization, where PDE parameters are observed to influence PINN performance, though without

theoretical analysis. In contrast, our work is the first to theoretically demonstrate that in singularly perturbed problems, smaller $\varepsilon$ values lead to greater training difficulty (**Theorem 1**). While both approaches share the curriculum-style motivation, our method differs in design and rigor: we construct a continuous homotopy path in parameter space with convergence guarantees and introduce a principled strategy for choosing the homotopy step size $\Delta\varepsilon$ (**Theorem 2**), which is absent in (Krishnapriyan et al., 2021).

**Contributions.** Our key contributions are summarized as follows:

- We propose *Homotopy Dynamics*, a novel method for solving singularly perturbed PDEs with neural networks, achieving improved training performance (Section 3).
- We provide a theoretical analysis of how the PDE parameter $\varepsilon$ affects training difficulty, and establish the convergence of our method (Section 4).
- We validate the method on diverse problems, including the Allen–Cahn equation, high-dimensional Helmholtz equation, and operator learning for Burgers' equation (Section 5).

## 2. Problem Setup

We begin by introducing the singularly perturbed problems studied in this work, followed by the neural network-based solution approach and the training challenges that motivate our method.

### 2.1. Singularly perturbed Problems

The form of the singularly perturbed problem is defined as follows:

$$\begin{cases} \mathcal{L}_\varepsilon u = f(u), & \text{in } \Omega, \\ \mathcal{B}u = g(x), & \text{on } \partial\Omega, \end{cases} \quad (1)$$

where $\mathcal{L}_\varepsilon$ is a differential operator defining the PDE with certain parameters, $\mathcal{B}$ is an operator associated with the boundary and/or initial conditions, and $\Omega \subseteq \mathbb{R}^d$. In the considered PDEs, the parameter $\varepsilon$ governs the complexity of the solution, with smaller values generally leading to more challenging behaviors. For example, in the Allen–Cahn equation (4), $\varepsilon$ represents the interfacial width parameter, where smaller $\varepsilon$ results in sharper transition layers. In the Burgers equation (28), $\varepsilon$ corresponds to the viscosity coefficient, with small values leading to steep gradients or shock-like structures. In the Helmholtz equation (27), $\varepsilon$ is inversely related to the wave number, and decreasing $\varepsilon$ yields higher-frequency oscillations.

In all cases, as $\varepsilon$ becomes small, the solution exhibits increased complexity—whether through sharp interfaces, steep gradients, or high-frequency structures—posing significant challenges for neural network-based solvers. More details will be provided in the following section.

## 2.2. Neural Networks for Solving PDEs

In this section, we focus on solution approximation rather than operator learning for simplicity, specifically using a neural network to approximate the PDE solution. In Section 5, we will demonstrate that our *Homotopy Dynamics* can also generalize to the operator learning case. The PDE problem is typically reformulated as the following non-linear least-squares problem, aiming to determine the parameters $\boldsymbol{\theta}$ of the neural network $u(x; \boldsymbol{\theta})$ (commonly a multi-layer perceptron, MLP):

$$
\min_{\boldsymbol{\theta} \in \mathbb{R}^p} L(\boldsymbol{\theta}) := \underbrace{\frac{1}{2n_{\text{res}}} \sum_{i=1}^{n_{\text{res}}} \left( \mathcal{L}_\varepsilon u(\boldsymbol{x}_r^i; \boldsymbol{\theta}) - f(u(\boldsymbol{x}_r^i; \theta)) \right)^2}_{L_{\text{res}}}
$$
$$
+ \lambda \underbrace{\frac{1}{2n_{\text{bc}}} \sum_{i=1}^{n_{\text{bc}}} \left( \mathcal{B}u(\boldsymbol{x}_b^j; \boldsymbol{\theta}) - g(\boldsymbol{x}_b^j) \right)^2}_{L_{\text{bc}}}. \quad (2)
$$

Here $L_{\text{res}}$ is the PDE residual loss, $L_{\text{bc}}$ is the boundary loss and $\lambda$ is a constant used to balance these two terms. The sets $\{\boldsymbol{x}_r^i\}_{i=1}^{n_{\text{res}}}$ represent represent the interior sample points, and $\{\boldsymbol{x}_b^j\}_{j=1}^{n_{\text{bc}}}$ represent the boundary sample points. We also introduce the $\ell_2$ relative error (L2RE) to evaluate the discrepancy between the neural network solution and the ground truth, defined as

$$
\text{L2RE} = \frac{\|u_{\boldsymbol{\theta}} - u^*\|_2}{\|u^*\|_2},
$$

where $u_{\boldsymbol{\theta}}$ is the neural network solution and $u^*$ is the ground truth.

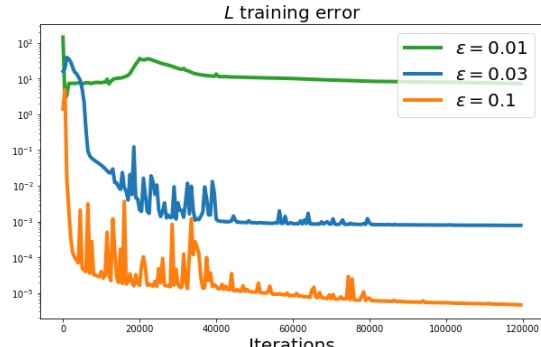

*Figure 2.* Training curves for different values of $\varepsilon$ in solving the 1D Allen-Cahn steady-state equation. As $\varepsilon$ decreases, the training error increases, indicating that the training process becomes progressively more difficult.

## 2.3. Challenges in Training Neural Networks

In this paper, we consider the following singularly perturbed elliptic problem:

$$
\begin{cases} -\varepsilon^2 \Delta u(x) = f(u), & x \in \Omega, \\ u(x) = g(x), & x \in \partial\Omega, \end{cases} \quad (3)
$$

where $\varepsilon$ is a problem parameter that influences both the structure of the solution and the difficulty of training neural network solvers. As a representative example, we focus on the steady-state Allen–Cahn equation in one spatial dimension:

$$
\begin{cases} -\varepsilon^2 u''(x) = u^3 - u, & x \in [0, 1], \\ u(0) = -1, \quad u(1) = 1, \end{cases} \quad (4)
$$

where the parameter $\varepsilon$ controls the width of the internal interface. As $\varepsilon$ decreases, the interface becomes increasingly sharp, resulting in a solution with higher gradients near the transition region. The analytic steady-state solution of this problem is given by

$$
u(x) = \tanh\left( \frac{x - 0.5}{\sqrt{2}\varepsilon} \right),
$$

where the interface is centered at $x = 0.5$. As shown in Figure 3, the solution becomes sharper as $\varepsilon$ becomes smaller.

To show the challenges in the optimization problem defined in (2), we present the training curves for varying values of $\varepsilon$ in Figure 2. As $\varepsilon$ decreases, training errors increase. This is due to the significantly increased training difficulty and slower convergence for smaller $\varepsilon$. In the subsequent sections, we analyze the underlying reasons for this phenomenon and introduce a homotopy dynamics-based approach to address the challenge.

## 3. Homotopy Dynamics

To address training difficulties in neural networks for singularly perturbed problems, we introduce a novel approach termed *homotopy dynamics*.

### 3.1. Homotopy Path Tracking

First, we introduce the homotopy function below:

$$H(u, \varepsilon) = \mathcal{L}_\varepsilon u - f(u) \equiv 0, \tag{5}$$

where $\varepsilon$ is the parameter in the PDEs. Specifically, this formulation represents the PDE problem $\mathcal{L}_\varepsilon u = f(u)$. In this context, $\varepsilon$ is treated as a path-tracking parameter. At $\varepsilon = \varepsilon_0$, we assume that the solutions to $H(u_0, \varepsilon_0) = 0$ are either known or can be easily approximated by neural networks. These solutions are referred to as the starting points. At $\varepsilon = \varepsilon^*$, the original system we aim to solve is recovered, which is referred to as the target system. Therefore, solving the target system involves tracking the solutions of $H(u, \varepsilon) = 0$ from $\varepsilon = \varepsilon_0$, where the solutions are known, to $\varepsilon = \varepsilon^*$, where the solutions are sought.

The process of path tracking between $\varepsilon_0$ and $\varepsilon^*$ is governed by solving the Davidenko differential equation:

$$\frac{\mathrm{d}H(u(\varepsilon), \varepsilon)}{\mathrm{d}\varepsilon} = \frac{\partial H(u(\varepsilon), \varepsilon)}{\partial u}\frac{\mathrm{d}u(\varepsilon)}{\mathrm{d}\varepsilon} + \frac{\partial H(u(\varepsilon), \varepsilon)}{\partial \varepsilon} = 0, \tag{6}$$

with the initial condition $u(\varepsilon_0) = u_0$. Thus, path tracking reduces to numerically solving an initial value problem, with the starting points acting as the initial conditions. Additionally, the boundary condition in (3) should be taken into account when solving the initial value problem numerically.

### 3.2. Incorporating Homotopy Dynamics into Neural Network Training

To enhance the neural network training process, we incorporate homotopy dynamics by gradually transitioning the network from an easier problem (with a larger $\varepsilon_0$) to the original target problem (with $\varepsilon^*$). This approach helps mitigate the challenges associated with training networks for problems involving small values of $\varepsilon$, where solutions become increasingly sharp or oscillation and harder to compute. Specifically, we denote the neural network solution for (3) as $u(x; \boldsymbol{\theta}(\varepsilon))$. The homotopy path tracking for training neural networks can then be refined as:

$$H_u \nabla_{\boldsymbol{\theta}} u \cdot \frac{\mathrm{d}\boldsymbol{\theta}(\varepsilon)}{\mathrm{d}\varepsilon} + H_\varepsilon = 0, \tag{7}$$

where $H_u = \frac{\partial H}{\partial u}$, $H_\varepsilon = \frac{\partial H}{\partial \varepsilon}$ and $\nabla_{\boldsymbol{\theta}} u$ represents the Jacobian with respect to the neural network parameters $\boldsymbol{\theta}$. Thus we can derive the homotopy dynamics system as:

$$\frac{\mathrm{d}\boldsymbol{\theta}(\varepsilon)}{\mathrm{d}\varepsilon} = -(H_u \nabla_{\boldsymbol{\theta}} u)^\dagger H_\varepsilon, \quad \varepsilon \in [\varepsilon_0, \varepsilon^*], \tag{8}$$

with the initial condition $\boldsymbol{\theta}(\varepsilon_0) = \boldsymbol{\theta}_0$ and $\dagger$ stands for Moore–Penrose inverse (Ben-Israel & Greville, 2006). Thus, to solve the singularly perturbed problem (3) where $\varepsilon$ is small, we can first solve (3) with a large $\varepsilon$ using the loss function (2). Then, by following the homotopy dynamics path tracking (8), we can progressively obtain the solution for smaller values of $\varepsilon$, ultimately solving the singularly perturbed problem.

In particular, path tracking in homotopy dynamics reduces to solving initial value problems numerically, with the start points serving as the initial conditions. For different neural network architectures, we propose two strategies, which are summarized in **Algorithm 1.**

One is to solve the initial value problem by using the forward Euler method, as follows:

$$\boldsymbol{\theta}(\varepsilon_k) = \boldsymbol{\theta}(\varepsilon_{k-1}) - \Delta\varepsilon_k \nabla_{\boldsymbol{\theta}} u(\varepsilon_{k-1})^\dagger H_u^{-1} H_\varepsilon, \tag{9}$$

where $\Delta\varepsilon_k = \varepsilon_k - \varepsilon_{k-1}$. This approach is effective for small neural networks, as the pseudo-inverse is easy to compute.

The other approach is to introduce the Homotopy Loss in the optimization, formulated as:

$$\min_{\boldsymbol{\theta}(\boldsymbol{\varepsilon_k}) \in \mathbb{R}^p} L_{\mathrm{Hom}}(\boldsymbol{\theta}(\varepsilon_k)) := L_H + \lambda L_{\mathrm{bc}} + \alpha L_{H_\varepsilon}, \tag{10}$$

where $L_H$ is defined in Eq. (12), and $L_{H_\varepsilon}$ is the loss function from Homotopy Dynamics, which is

$$L_{H_\varepsilon} = \frac{1}{2n_{\mathrm{res}}} \sum_{i=1}^{n_{\mathrm{res}}} \left( H_u(u_{\boldsymbol{\theta}(\varepsilon_k)}(\boldsymbol{x}_r^i), \varepsilon)\frac{\Delta u_k}{\Delta \varepsilon_k} + H_\varepsilon(u_{\boldsymbol{\theta}}(\boldsymbol{x}_r^i), \varepsilon) \right)^2.$$

This approach is suitable for large neural networks, as it does not require the computation of the pseudo-inverse, and $\Delta u_k = u_{\boldsymbol{\theta}(\varepsilon_k)} - u_{\boldsymbol{\theta}(\varepsilon_{k-1})}$.

---

**Algorithm 1** Homotopy Dynamics Path Tracking

**input** tolerance $\tau$, list of parameters $\varepsilon_0, \varepsilon_1, \ldots, \varepsilon_n$
   **Phase I: Directly train NN for large $\varepsilon_0$**
   **while** $L(\boldsymbol{\theta}(\varepsilon_0)) > \tau$ **do**
     $\min L(\boldsymbol{\theta}(\varepsilon_0))$
   **end while**
   **Phase II: Homotopy dynamics path tracking**
   **for** $k = 1, \ldots, n$ **do**
     $\Delta\varepsilon_k = \varepsilon_k - \varepsilon_{k-1}$

     **Strategy 1. Numerical solution via Forward Euler:**
     $\boldsymbol{\theta}(\varepsilon_k) = \boldsymbol{\theta}(\varepsilon_{k-1}) - \Delta\varepsilon_k \nabla_{\boldsymbol{\theta}} u(\varepsilon_{k-1})^\dagger H_u^{-1} H_\varepsilon$

     **Strategy 2. Optimization using homotopy loss:**
     **while** $L_{\mathrm{Hom}}(\boldsymbol{\theta}(\varepsilon_k)) > \tau$ **do**
     $\min L_{\mathrm{Hom}}(\boldsymbol{\theta}(\varepsilon_k))$
     **end while**

   **end for**
**output** $u_{\boldsymbol{\theta}(\varepsilon_n)}$

---

If $L_H$ and $\lambda L_{bc}$ are omitted in strategy 2, then strategy 2 can be viewed as an alternative approach to solving the linear

*Table 1.* Training loss and $L^2$ error (L2RE) for classical training vs. homotopy dynamics under different $\varepsilon$. Homotopy dynamics achieves consistently lower loss and error.

| | $\varepsilon = 0.1$ | | $\varepsilon = 0.03$ | | $\varepsilon = 0.01$ | |
|---|---|---|---|---|---|---|
| | Loss | L2RE | Loss | L2RE | Loss | L2RE |
| Classical | 5.00e-6 | 1.71e-2 | 7.76e-4 | 1.11 | 7.21 | 8.17e-1 |
| Homotopy | 5.00e-6 | 1.71e-2 | 7.45e-8 | 9.83e-3 | **4.63e-8** | **8.08e-3** |

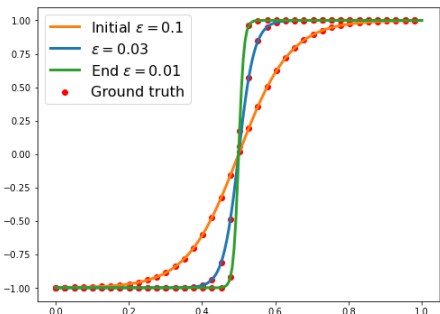

*Figure 3.* Evolution of the Homotopy dynamics for steady state 1D Allen-Cahn equation. The L2RE for $\varepsilon = 0.01$ is $8.08e - 3$.

system given in Eq. (7). However, for certain PDEs or larger neural networks, directly solving Eq. (7) is unstable because the term $H_u \nabla_\theta u$ contains many small singular values, causing conventional methods (e.g., using SVD) to incur large errors. Therefore, we opt to solve an optimization problem in the traditional manner. Within this framework, strategy 2 follows the same dynamic process as strategy 1, yielding more stable training. Including $L_H$ and $\lambda L_{bc}$ ensures that the obtained solution satisfies the target PDE. Conversely, even if $H(u, \varepsilon) = \text{Const} \neq 0$, the solutions still follow the same homotopy dynamics since they share the same $L_{H_\varepsilon}$.

**Example: 1D Allen-Cahn steady-state equation.** We demonstrate our proposed method on the one-dimensional Allen-Cahn steady-state equation by defining the following homotopy function:

$$H(u_\theta, \varepsilon) = \varepsilon^2 u_\theta''(x) + u_\theta^3 - u_\theta \equiv 0. \quad (11)$$

Following the homotopy dynamics in Eq. (8), we set the initial value at $\varepsilon = 0.1$ and gradually decrease it to the final value $\varepsilon_n = 0.01$. The initial solution, $\theta(\varepsilon_0)$, is obtained using the standard training process by directly minimizing (2). The results and the evolution process are presented in Table 1 and Figure 3. These results show that when $\varepsilon$ is large, the original training method achieves a relatively small error, leading to an accurate solution. However, as $\varepsilon$ decreases, the error increases, which reduces the accuracy of the solution. In contrast, the homotopy dynamics-based approach maintains accuracy effectively as $\varepsilon$ decreases.

## 4. Theoretical analysis

In this section, we provide theoretical support for homotopy dynamics. In the first part, we demonstrate that for certain PDEs with small parameters, direct training using PINN methods is highly challenging. This analysis is based on the neural tangent kernel (NTK) framework (Allen-Zhu et al., 2019). In the second part, we show that homotopy dynamics will converge to the solution with a small parameter $\varepsilon$, provided that the dynamic step size is sufficiently small and the initial solution has been well learned by the neural network.

### 4.1. Challenges in Training Neural Network with Small Certain Parameters

Let us consider training neural networks without homotopy dynamics. The corresponding loss function can be expressed as

$$L_H(\boldsymbol{\theta}) = \frac{1}{2n} \sum_{i=1}^{n} H^2(u_{\boldsymbol{\theta}}(\boldsymbol{x}_i), \varepsilon), \quad (12)$$

where $\{\boldsymbol{x}_i\}_{i=1}^{n}$ represents the training data used to optimize the neural network. Here, we assume that the parameter $\varepsilon$ in the PDE appears only in the interior terms and not in the boundary conditions. Therefore, in this section, we omit the effect of boundary conditions, as the behavior at the boundary remains unchanged for any given $\varepsilon$.

Furthermore, to simplify the notation, we use $n$ instead of $n_{\text{res}}$ and denote $\boldsymbol{x}_r^i$ simply as $\boldsymbol{x}_i$ comparing with Eq. (2).

In the classical approach, such a loss function is optimized using gradient descent, stochastic gradient descent, or Adam. Considering the training process of gradient descent in its continuous form, it can be expressed as:

$$\begin{aligned} \frac{\mathrm{d}\boldsymbol{\theta}}{\mathrm{d}t} &= -\nabla_{\boldsymbol{\theta}} L_H(\boldsymbol{\theta}) \\ &= -\frac{1}{n} \sum_{i=1}^{n} H(u_{\boldsymbol{\theta}}(\boldsymbol{x}_i), \varepsilon) \delta_{u_{\boldsymbol{\theta}}} H(u_{\boldsymbol{\theta}}(\boldsymbol{x}_i), \varepsilon) \nabla_{\boldsymbol{\theta}} u_{\boldsymbol{\theta}}(\boldsymbol{x}_i), \\ &= -\frac{1}{n} \boldsymbol{H}(u_{\boldsymbol{\theta}}(\boldsymbol{x}), \varepsilon) \cdot \boldsymbol{S}, \quad (13) \end{aligned}$$

where $t$ in this section is the time of the gradient descent flow, $\delta_{u_{\boldsymbol{\theta}}}$ is the functional variational corresponding to $u_{\boldsymbol{\theta}}$, and

$$\begin{aligned} \boldsymbol{H}(u_{\boldsymbol{\theta}}(\boldsymbol{x}), \varepsilon) &:= \left[ H(u_{\boldsymbol{\theta}}(\boldsymbol{x}_i), \varepsilon) \delta_{u_{\boldsymbol{\theta}}} H(u_{\boldsymbol{\theta}}(\boldsymbol{x}_i), \varepsilon) \right]_{i=1}^{n} \\ &= \boldsymbol{l}_\varepsilon \cdot \boldsymbol{D}_\varepsilon, \quad (14) \end{aligned}$$

and

$$\boldsymbol{l}_\varepsilon := \left[ H(u_{\boldsymbol{\theta}}(\boldsymbol{x}_i, \boldsymbol{\theta}), \varepsilon) \right]_{i=1}^{n} \in \mathbb{R}^{1 \times n}, \boldsymbol{D}_\varepsilon \in \mathbb{R}^{n \times n} \quad (15)$$

where $\boldsymbol{D}_\varepsilon$ represents the discrete form of the variation of PDEs in different scenarios. Furthermore,

$$\boldsymbol{S} = \left[ \nabla_{\boldsymbol{\theta}} u_{\boldsymbol{\theta}}(\boldsymbol{x}_1), \dots, \nabla_{\boldsymbol{\theta}} u_{\boldsymbol{\theta}}(\boldsymbol{x}_n) \right]. \quad (16)$$

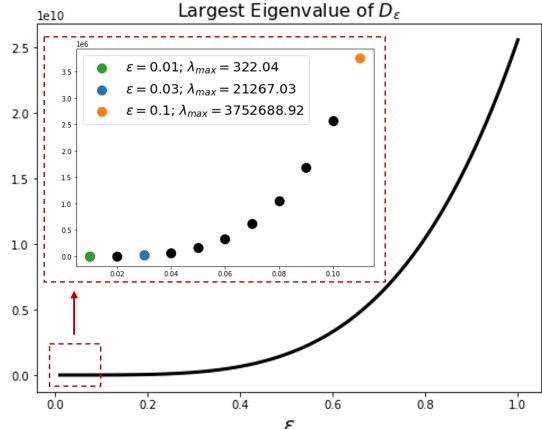

*Figure 4.* Largest eigenvalue of $\boldsymbol{D}_\varepsilon$ (21) for different $\varepsilon$. A smaller $\varepsilon$ results in a smaller largest eigenvalue of (21), leading to a slower convergence rate and increased difficulty in training.

Therefore, we obtain

$$
\begin{aligned}
\frac{\mathrm{d}L_H(\boldsymbol{\theta})}{\mathrm{d}t} &= \nabla_{\boldsymbol{\theta}} L_H(\boldsymbol{\theta}) \frac{\mathrm{d}\boldsymbol{\theta}}{\mathrm{d}t} \\
&= -\frac{1}{n^2} \boldsymbol{H}(u_{\boldsymbol{\theta}}(\boldsymbol{x}), \varepsilon) \boldsymbol{S}\boldsymbol{S}^\top \boldsymbol{H}^\top(u_{\boldsymbol{\theta}}(\boldsymbol{x}), \varepsilon) \\
&= -\frac{1}{n^2} \boldsymbol{l}_\varepsilon \boldsymbol{D}_\varepsilon \boldsymbol{S}\boldsymbol{S}^\top \boldsymbol{D}_\varepsilon^\top \boldsymbol{l}_\varepsilon^\top.
\end{aligned} \tag{17}
$$

Hence, the kernel of the gradient descent update is given by

$$
\boldsymbol{K}_\varepsilon := \boldsymbol{D}_\varepsilon \boldsymbol{S}\boldsymbol{S}^\top \boldsymbol{D}_\varepsilon^\top. \tag{18}
$$

The following theorem provides an upper bound for the smallest eigenvalue of the kernel and its role in the gradient descent dynamics:

**Theorem 4.1** (Effectiveness of Training via the Eigenvalue of the Kernel). *Suppose $\lambda_{\min}(\boldsymbol{S}\boldsymbol{S}^\top) > 0$ and $\boldsymbol{D}_\varepsilon$ is non-singular, and let $\varepsilon \geq 0$ be a constant. Then, we have $\lambda_{\min}(\boldsymbol{K}_\varepsilon) > 0$, and there exists $T > 0$ such that*

$$
L_H(\boldsymbol{\theta}(t)) \leq L_H(\boldsymbol{\theta}(0)) \exp\left(-\frac{\lambda_{\min}(\boldsymbol{K}_\varepsilon)}{n} t\right) \tag{19}
$$

*for all $t \in [0, T]$. Furthermore,*

$$
\begin{aligned}
\lambda_{\min}(\boldsymbol{S}\boldsymbol{S}^\top) &\lambda_{\min}(\boldsymbol{D}_\varepsilon \boldsymbol{D}_\varepsilon^\top) \\
&\leq \lambda_{\min}(\boldsymbol{K}_\varepsilon) \leq \lambda_{\min}(\boldsymbol{S}\boldsymbol{S}^\top) \lambda_{\max}(\boldsymbol{D}_\varepsilon \boldsymbol{D}_\varepsilon^\top). \tag{20}
\end{aligned}
$$

*Remark* 4.2. For $\boldsymbol{S}\boldsymbol{S}^\top$, previous works such as (Luo & Yang, 2020; Allen-Zhu et al., 2019; Arora et al., 2019; Cao & Gu, 2020; Yang et al., 2025; Du et al., 2019; Li et al., 2020) demonstrate that it becomes positive when the width of the neural network is sufficiently large with ReLU activation functions or smooth functions. Additionally, (Gao

et al., 2023) discusses the positivity of the gradient kernel in PINNs for solving heat equations. Therefore, we can reasonably assume that $\boldsymbol{S}\boldsymbol{S}^\top$ is a strictly positive matrix. In Appendix A.1, we present a specific scenario where $\lambda_{\min}(\boldsymbol{S}\boldsymbol{S}^\top) > 0$ holds with high probability.

This theorem demonstrates that the smallest eigenvalue of the kernel directly affects the training speed. Equation (20) shows that the upper bound of $\lambda_{\min}(\boldsymbol{K}_\varepsilon)$ can be influenced by $\lambda_{\max}(\boldsymbol{D}_\varepsilon \boldsymbol{D}_\varepsilon^\top)$. In many PDE settings, the maximum eigenvalue $\lambda_{\max}(\boldsymbol{D}_\varepsilon \boldsymbol{D}_\varepsilon^\top)$ tends to be small when $\varepsilon$ is small. For example, in this paper, we consider the Allen–Cahn equation, given by

$$
-\varepsilon^2 \Delta u + f(u) = 0,
$$

where $f(u) = u^3 - u$. In this case, $\boldsymbol{D}_\varepsilon$ corresponds to the discrete form of the operator $-\varepsilon^2 \Delta + f'(u)$, which can be written as

$$
\boldsymbol{D}_\varepsilon = -\varepsilon^2 \Delta_{\mathrm{dis}} + \mathrm{diag}\big(f'(u(\boldsymbol{x}_1)), \ldots, f'(u(\boldsymbol{x}_n))\big). \tag{21}
$$

According to (Morton & Mayers, 2005), the discrete Laplacian $-\varepsilon^2 \Delta_{\mathrm{dis}}$ is strictly positive. Specifically, in the one-dimensional case, its largest eigenvalue is given by

$$
4\varepsilon^2 n^2 \cos^2 \frac{\pi}{2n+1},
$$

which is close $4\varepsilon^2 n^2$ as $n$ is large enough.

Moreover, since $f'(u(\boldsymbol{x}_i))$ lies within the interval $[-1, 2]$, when $\varepsilon$ is large (i.e., close to 1), the largest eigenvalue of $\boldsymbol{D}_\varepsilon$ becomes very large regardless of the sampling locations $\{\boldsymbol{x}_i\}_{i=1}^n$ (see Figure 4 for the case $n = 200$. Other equations exhibit similar behavior. Please refer to the Appendix B for detail.) Consequently, by Theorem 4.1 the upper bound on the smallest eigenvalue of $\boldsymbol{K}_\varepsilon$ also becomes large—specifically, it is of order $n^4$ in this case due to Weyl's inequalities. As a result, the training speed can achieve a rate of $\exp(-Cn^3 t)$ (see Eq. (19)), which is rapid and indicates that training is relatively easy. In contrast, the lower bound for the training speed is given by $\exp(-Ct/n)$.

The fastest rate is attained in the special situation where there exists a nonzero vector $\boldsymbol{x}$ that is an eigenvector corresponding to the largest eigenvalue of $\boldsymbol{D}_\varepsilon^\top \boldsymbol{D}_\varepsilon$ and, simultaneously, $\boldsymbol{D}_\varepsilon \boldsymbol{x}$ is an eigenvector corresponding to the smallest eigenvalue of $\boldsymbol{S}^\top \boldsymbol{S}$. This scenario may occur under particular configurations of $\boldsymbol{S}$ and $\boldsymbol{D}_\varepsilon$, which in turn depend on the underlying PDE and the distribution of the sampling points.

On the other hand, when $\varepsilon$ is small (close to 0), the largest eigenvalue of $\boldsymbol{D}_\varepsilon$ is only of order 1 with respect to $n$. Consequently, the upper bound on the smallest eigenvalue of $\boldsymbol{K}_\varepsilon$ no longer scales as a constant with respect to $n$. In this case, the training speed is reduced to $\exp(-Ct/n)$ (as indicated in Eq. (19)), which is significantly slower and suggests that

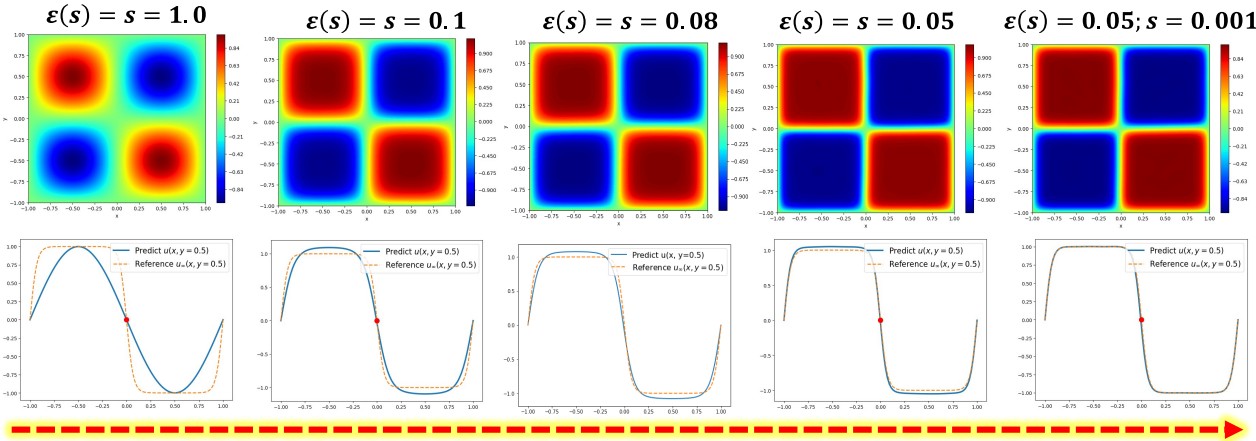

*Figure 5.* 2D Allen Cahn Equaiton. (Top) Evolution of the Homotopy Dynamcis. (Bottom) Plot for Cross-section of $u(x, y)$ at $y = 0.5$ i.e., $u(x, y = 0.5)$. The reference solution $u_\infty(x)$ represents the ground truth steady-state solution. The L2RE is $8.78e-3$. Number of residual points is $n_{\text{res}} = 50 \times 50$.

training becomes difficult. Therefore, while a larger $\varepsilon$ may yield relatively easy training in some instances, a smaller $\varepsilon$ will invariably lead to challenging training conditions.

### 4.2. Convergence of Homotopy Dynamics

In this section, we aim to demonstrate that homotopy dynamics is a reasonable approach for obtaining the solution when $\varepsilon$ is small. Recall that Strategy 2 is merely an alternative approach for solving the linear system in our framework, with the underlying principles remaining the same. Therefore, our theoretical analysis is primarily based on Strategy 1. For simplicity of notation, we denote $u(\varepsilon)$ as the exact solution of $H(u, \varepsilon) = 0$ and $U(\varepsilon)$ as its numerical approximation in the simulation. Suppose $H(u(\varepsilon), \varepsilon) = 0$, and assume that $\frac{\partial H(u(\varepsilon), \varepsilon)}{\partial u}$ is invertible. Then, the dynamical system (6) can be rewritten as

$$\frac{du}{d\varepsilon} = -\left(\frac{\partial H(u(\varepsilon), \varepsilon)}{\partial u}\right)^{-1} \frac{\partial H(u(\varepsilon), \varepsilon)}{\partial \varepsilon} =: h(u(\varepsilon), \varepsilon). \tag{22}$$

Applying Euler's method to this dynamical system, we obtain

$$U(\varepsilon_{k+1}) = U(\varepsilon_k) + (\varepsilon_{k+1} - \varepsilon_k)h(U(\varepsilon_k), \varepsilon_k). \tag{23}$$

The following theorem shows that if $u(\varepsilon_0) - U(\varepsilon_0)$ is small and the step size $(\varepsilon_{k+1} - \varepsilon_k)$ is sufficiently small at each step, then $u(\varepsilon_k) - U(\varepsilon_k)$ remains small.

**Theorem 4.3** (Convergence of Homotopy Dynamics). *Suppose $h(\varepsilon, u)$ is a continuous operator for $0 < \varepsilon_n \le \varepsilon_0$ and $u \in H^2(\Omega)$, and*

$$\|h(u_1, \varepsilon) - h(u_2, \varepsilon)\|_{H^2(\Omega)} \le P_\varepsilon \|u_1 - u_2\|_{H^2(\Omega)}.$$

*Assume there exists a constant $P$ such that $(\varepsilon_k - \varepsilon_{k+1})P_{\varepsilon_k} \le P \cdot \frac{\varepsilon_0 - \varepsilon_n}{n}$, $e_0 := \|u(\varepsilon_0) - U(\varepsilon_0)\|_{H^2(\Omega)} \ll 1$ and*

$$\tau := \frac{n}{\varepsilon_0 - \varepsilon_n} \sup_{0 \le k \le n} (\varepsilon_k - \varepsilon_{k+1})^2 \|u(\varepsilon_k)\|_{H^4(\Omega)} \ll 1,$$

*then we have*

$$\|u(\varepsilon_n) - U(\varepsilon_n)\|_{H^2(\Omega)}$$
$$\le e_0 e^{P(\varepsilon_0 - \varepsilon_n)} + \frac{\tau(e^{P(\varepsilon_0 - \varepsilon_n)} - 1)}{2P} \ll 1. \tag{24}$$

The proof of Theorem 4.3 is inspired by (Atkinson et al., 2009).

Theorem 4.3 shows that if $e_0$ is small and the step size $(\varepsilon_{k+1} - \varepsilon_k)$ is sufficiently small at each step and satisfies $(\varepsilon_k - \varepsilon_{k+1})P_{\varepsilon_k} \le P \cdot \frac{\varepsilon_0 - \varepsilon_n}{n}$ i.e., the training step size should depend on the Lipschitz constant of $h(u, \varepsilon)$, ensuring stable training, then $u(\varepsilon_k) - U(\varepsilon_k)$ remains small. In other words, when $\varepsilon \to 0$, $P_\varepsilon$ may not be bounded. Nonetheless, we do not require $\varepsilon$ to be exactly zero; it only needs to be a small constant. For a small $\varepsilon$, $P_\varepsilon$ might be large but remains finite. In this case, one must choose sufficiently small steps $\varepsilon_{k+1} - \varepsilon_k$ to ensure that the training error stays controlled. The initial error $e_0$ can be very small since we use a neural network to approximate the solution of PDEs for large $\varepsilon$, where learning is effective.

The total error $e_0$ consists of approximation, generalization, and training errors. The training error can be effectively controlled when $\varepsilon$ is large (Theorem 4.1), while the approximation and generalization errors remain small if the sample size is sufficiently large and the neural network is expressive enough. Theoretical justifications are provided in (Yang

*Table 2.* $\lambda_{\min}(\boldsymbol{K}_\varepsilon)$ for different initialization for $\varepsilon = 0.01$ in Equation (4).

| Initialization | Xavier | Hom $\varepsilon = 0.1$ | Hom $\varepsilon = 0.05$ | Hom $\varepsilon = 0.03$ | Hom $\varepsilon = 0.02$ |
|---|---|---|---|---|---|
| $\lambda_{\min}(\boldsymbol{K}_\varepsilon)$ | 7.38e-8 | 2.11e-6 | 7.77e-5 | 1.57e-4 | 1.48e-2 |

*Table 3.* **2D Allen-Cahn Equation.** Relative $L^2$ error comparison across various training strategies.

| Method | Original PINN | Curriculum (Krishnapriyan et al., 2021) | Time Seq. (Wight & Zhao, 2020; Mattey & Ghosh, 2022) | Resampling | **Homotopy** |
|---|---|---|---|---|---|
| L2RE | 9.56e-1 | 8.89e-1 | 8.95e-2 | 8.25e-1 | **8.78e-3** |

et al., 2023; Yang & He, 2024) and further discussed in Appendix A.4.

For Strategy 2, we enforce $H_\varepsilon = 0$ by retraining the network from scratch at each homotopy step rather than by integrating the path with Euler's method. Each iteration uses the previous solution as the initial guess for the next, producing progressively better starting points. To illustrate, we consider the 1D Allen–Cahn equation (Eq. (4)) with $\varepsilon = 0.01$. As shown in Table 2, smaller increments $\Delta\varepsilon$ increase $\lambda_{\min}(\boldsymbol{K}_\varepsilon)$, indicating improved conditioning and faster convergence. In our experiments, $\varepsilon = 0.02$ yielded the best initialization for $\varepsilon = 0.01$, highlighting that selecting a homotopy parameter close to the final target provides the most effective initial guess—this encapsulates the key idea behind Strategy 2.

## 5. Experiments

We conduct several experiments across different problem settings to assess the efficiency of our proposed method. In the following experiments, we adopt Strategy 2 for all training procedures. Detailed descriptions of the experimental settings are provided in Appendix B.

### 5.1. 2D Allen Cahn Equation

First, we consider the following time-dependent problem:

$$
\begin{aligned}
&u_t = \varepsilon^2 \Delta u - u(u^2 - 1), \quad (x, y) \in [-1, 1] \times [-1, 1] \\
&u(x, y, 0) = -\sin(\pi x)\sin(\pi y) \\
&u(-1, y, t) = u(1, y, t) = u(x, -1, t) = u(x, 1, t) = 0.
\end{aligned} \tag{25}
$$

We aim to find the steady-state solution for this equation with $\varepsilon = 0.05$ and define the homotopy as:

$$
H(u, s, \varepsilon) = (1 - s)\left(\varepsilon(s)^2 \Delta u - u(u^2 - 1)\right) + s(u - u_0),
$$

where $s \in [0, 1]$. Specifically, when $s = 1$, the initial condition $u_0$ is automatically satisfied, and when $s = 0$, it recovers the steady-state problem. The function $\varepsilon(s)$ is given by

$$
\varepsilon(s) = \begin{cases} s, & s \in [0.05, 1], \\ 0.05, & s \in [0, 0.05]. \end{cases} \tag{26}
$$

Here, $\varepsilon(s)$ varies with $s$ during the first half of the evolution. Once $\varepsilon(s)$ reaches 0.05, it remains fixed, and only $s$ contin-

ues to evolve toward 0. As shown in Figure 5 and Table 3, our method achieves superior solution accuracy compared to the other approaches.

### 5.2. High-Dimensional Helmholtz Equation

One of the advantages of solving differential equations using neural networks is their potential to overcome the curse of dimensionality and tackle high-dimensional problems. In this example, we demonstrate this capability by comparing the performance of the standard PINN approach with our proposed homotopy-based training method on the following high-dimensional Helmholtz equation:

$$
-\varepsilon^2 \Delta u - \tfrac{1}{d} u = 0 \quad \text{in } \Omega, \qquad u = g \quad \text{on } \partial\Omega, \tag{27}
$$

where $\Omega = [-1, 1]^d$. This problem admits the exact solution

$$
u(\boldsymbol{x}) = \sin\left(\frac{1}{d} \sum_{i=1}^{d} \frac{1}{\varepsilon} x_i\right).
$$

The corresponding homotopy is defined by

$$
H(u, \varepsilon) = \varepsilon^2 \Delta u + \frac{1}{d} u.
$$

Here, we start the homotopy training by $\varepsilon_0 = 1$. The numerical results for dimension $d = 20$ are reported in Table 4, where we compare the relative $L^2$ errors obtained by classical PINN training and the proposed homotopy dynamics for different values of $\varepsilon$. As shown in the table, the homotopy-based approach achieves consistently lower errors, especially for small $\varepsilon$, where classical training suffers from significant accuracy degradation.

*Table 4.* Comparison of relative $L^2$ errors achieved by classical training and homotopy dynamics for different $\varepsilon$ values in the high-dimensional Helmholtz equation.

| Dimension $d = 20$ | $\varepsilon = 1/2$ | $\varepsilon = 1/20$ | $\varepsilon = 1/50$ |
|---|---|---|---|
| Classical Training | 1.23e-3 | 7.21e-2 | 9.98e-1 |
| Homotopy Dynamics | 5.86e-4 | 5.00e-4 | 5.89e-4 |

### 5.3. Burgers Equation

In this example, we adopt the operator learning framework to solve for the steady-state solution of the Burgers equation,

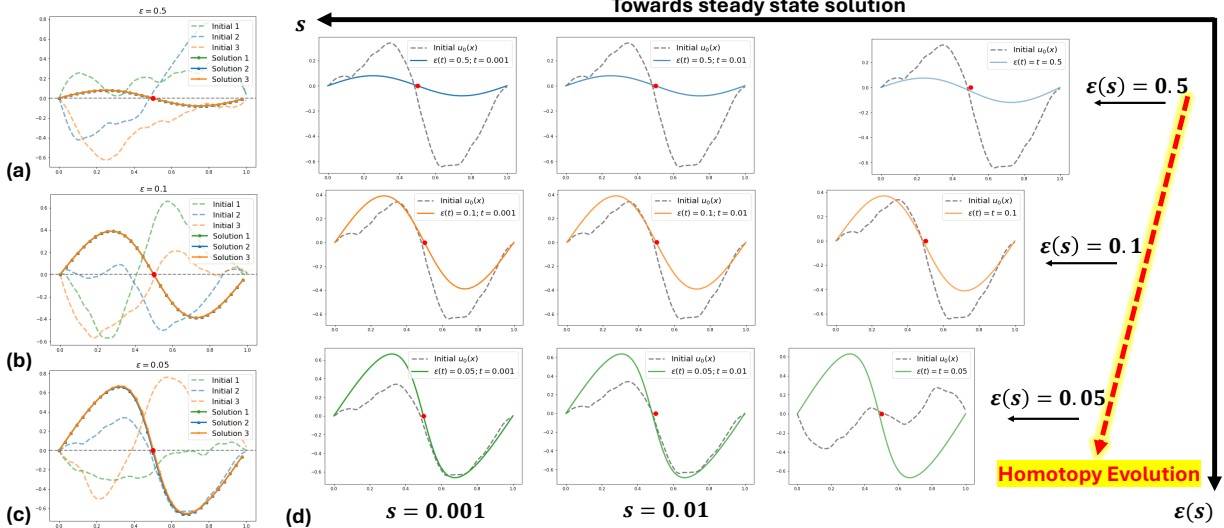

*Figure 6.* 1D Burgers' Equation (Operator Learning): Steady-state solutions for different initializations $u_0$ under varying viscosity $\varepsilon$: (a) $\varepsilon = 0.5$, (b) $\varepsilon = 0.1$, (c) $\varepsilon = 0.05$. The results demonstrate that all final test solutions converge to the correct steady-state solution. (d) Illustration of the evolution of a test initialization $u_0$ following homotopy dynamics. The number of residual points is $n_{\mathrm{res}} = 128$.

given by:

$$u_t + \left(\frac{u^2}{2}\right)_x - \varepsilon u_{xx} = \pi \sin(\pi x) \cos(\pi x), \quad x \in [0,1]$$
$$u(x,0) = u_0(x), \tag{28}$$
$$u(0,t) = u(1,t) = 0,$$

with Dirichlet boundary conditions, where $u_0 \in L_0^2((0,1);\mathbb{R})$ is the initial condition and $\varepsilon \in \mathbb{R}$ is the viscosity coefficient. We aim to learn the operator mapping the initial condition to the steady-state solution, $G^\dagger : L_0^2((0,1);\mathbb{R}) \to H_0^r((0,1);\mathbb{R})$, defined by $u_0 \mapsto u_\infty$ for any $r > 0$. As shown in Theorem 2.2 of (Kreiss & Kreiss, 1986) and Theorems 2.5 and 2.7 of (Hao & Yang, 2019), for any $\varepsilon > 0$, the steady-state solution is independent of the initial condition, with a single shock occurring at $x_s = 0.5$. Here, we use DeepONet (Lu et al., 2021a) as the network architecture. The homotopy definition, similar to Equation (25), can be found in Appendix B.6. The results can be found in Figure 6 and Table 5. Experimental results show that the homotopy dynamics strategy performs well in the operator learning setting as well. As shown in Appendix 8, DeepONet trained via homotopy dynamics achieves comparable accuracy but significantly faster inference than the finite difference method.

## 6. Conclusion

In this work, we explore the challenges of using neural networks to solve singularly perturbed problems. Specifically,

*Table 5.* Homotopy Dynamics Results on operator learning for Burgers Equation: Homotopy Loss, Relative $L^2$ Error, and Shock Localization Accuracy

| | $\varepsilon = 0.5$ | $\varepsilon = 0.1$ | $\varepsilon = 0.05$ |
|---|---|---|---|
| Homotopy Loss $L_H$ | 7.55e-7 | 3.40e-7 | 7.77e-7 |
| L2RE | 1.50e-3 | 7.00e-4 | 2.52e-2 |
| MSE Distance at $x_s$ | 1.75e-8 | 9.14e-8 | 1.2e-3 |

we analyze the training difficulties caused by certain parameters in the PDEs. To overcome these challenges, we propose a training method based on homotopy dynamics to avoid training original and singularly perturbed problems directly and to improve the training performance of neural networks to solve such problems. Our theoretical analysis supports the convergence of the proposed homotopy dynamics. Experimental results demonstrate that our method performs well across a range of singularly perturbed problems. In solution approximation tasks, it accurately captures the steady-state solutions of the Allen–Cahn equation and effectively handles high-dimensional Helmholtz equations with large wave numbers. Moreover, in the context of operator learning, it achieves strong performance on the Burgers' equation. Both the theoretical analysis and experimental results consistently validate the effectiveness of our proposed method.

Looking ahead, it will be valuable to explore whether homotopy dynamics can be applied to a wider range of practical problems. We believe that homotopy dynamics offers a natural entry point for comparing neural-network methods with traditional techniques.

## Acknowledgements

Y.Y. and W.H. was supported by National Institute of General Medical Sciences through grant 1R35GM146894. The work of Y.X. was supported by the Project of Hetao Shenzhen-HKUST Innovation Cooperation Zone HZQB-KCZYB-2020083.

We would like to acknowledge helpful comments from the anonymous reviewers and area chairs, which have improved this submission.

## Impact Statement

This paper presents work whose goal is to advance the field of scientific machine learning. There are many potential societal consequences of our work, none which we feel must be specifically highlighted here.

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

# A. Proofs of Theorems 4.1 and 4.3

Before we prove Theorem 4.1, note that our method can be readily generalized to deep neural networks, as the underlying theoretical techniques remain the same; we merely need to combine our approach with the results in (Du et al., 2019). In this paper, we employ a two-layer neural network to simplify the notation and enhance readability, focusing on explaining why training becomes challenging when $\varepsilon$ is small.

## A.1. $\lambda_{\mathbf{min}}(\boldsymbol{S}\boldsymbol{S}^\top) > 0$

In this subsection, we consider a two-layer neural network defined as follows:

$$\phi(\boldsymbol{x}; \boldsymbol{\theta}) := \frac{1}{\sqrt{m}} \sum_{k=1}^{m} a_k\, \sigma(\boldsymbol{\omega}_k^\top \boldsymbol{x}), \tag{29}$$

where the activation function $\sigma$ satisfies the following assumption:

**Assumption A.1.** The function $\sigma(\cdot)$ is analytic and not a polynomial. Moreover, there exists a positive constant $c$ such that

$$|\sigma(x)| \le c|x|$$

for all $x$.

Note that both $\sigma(x) = \ln(1 + \exp(x))$ and $\sigma(x) = \frac{1}{1+\exp(-x)}$ satisfy Assumption A.1.

We assume that the weights and biases are sampled as follows:

$$\boldsymbol{\omega}_k \sim N\left(0, \boldsymbol{I}_d\right), \quad a_k \sim N(0, 1), \tag{30}$$

where $N(0, 1)$ denotes the standard Gaussian distribution.

The kernels characterizing the training dynamics take the following form:

$$\begin{aligned} k^{[a]}(\boldsymbol{x}, \boldsymbol{x}') &:= \mathbf{E}_{\boldsymbol{\omega}} \sigma(\boldsymbol{\omega}^\top \boldsymbol{x}) \sigma(\boldsymbol{\omega}^\top \boldsymbol{x}') \\ k^{[\boldsymbol{\omega}]}(\boldsymbol{x}, \boldsymbol{x}') &:= \mathbf{E}_{(a,\boldsymbol{\omega})} a^2 \sigma'(\boldsymbol{\omega}^\top \boldsymbol{x}) \sigma'(\boldsymbol{\omega}^\top \boldsymbol{x}') \boldsymbol{x} \cdot \boldsymbol{x}'. \end{aligned} \tag{31}$$

The Gram matrices, denoted as $\boldsymbol{K}^{[a]}$ and $\boldsymbol{K}^{[\boldsymbol{\omega}]}$, corresponding to an infinite-width two-layer network with the activation function $\sigma$, can be expressed as follows:

$$\begin{aligned} K_{ij}^{[a]} &= k^{[a]}(\boldsymbol{x}_i, \boldsymbol{x}_j), \ \boldsymbol{K}^{[a]} = (K_{ij}^{[a]})_{n \times n}, \\ K_{ij}^{[\boldsymbol{\omega}]} &= k^{[\boldsymbol{\omega}]}(\boldsymbol{x}_i, \boldsymbol{x}_j), \ \boldsymbol{K}^{[\boldsymbol{\omega}]} = (K_{ij}^{[\boldsymbol{\omega}]})_{n \times n}. \end{aligned} \tag{32}$$

**Lemma A.2** ((Du et al., 2019)). *Suppose that Assumption A.1 holds and for any $i, j \in [n], i \ne j, \boldsymbol{x}_i \nparallel \boldsymbol{x}_j$. Then the matrices $\boldsymbol{K}^{[\boldsymbol{\omega}]}$ and $\boldsymbol{K}^{[a]}$ are strictly positive, i.e.*

$$\lambda_1 := \min\left\{\lambda_{min}\left(\boldsymbol{K}^{[\boldsymbol{\omega}]}\right), \lambda_{min}\left(\boldsymbol{K}^{[a]}\right)\right\} > 0. \tag{33}$$

It is easy to check that

$$\boldsymbol{K}^{[\boldsymbol{\omega}]} + \boldsymbol{K}^{[a]} = \lim_{m \to \infty} \boldsymbol{S}\boldsymbol{S}^\top \tag{34}$$

based on the law of large numbers. Furthermore, we can show that the accuracy decreases exponentially as the width of the neural network increases.

**Definition A.3** ((Vershynin, 2018)). A random variable $X$ is sub-exponential if and only if its sub-exponential norm is finite i.e.

$$\|X\|_{\psi_1} := \inf\{s > 0 \mid \mathbf{E}_X[e^{|X|/s} \le 2.] \tag{35}$$

Furthermore, the chi-square random variable $X$ is a sub-exponential random variable and $C_{\psi,d} := \|X\|_{\psi_1}$.

**Lemma A.4.** *Suppose that $\boldsymbol{w} \sim N\left(0, \boldsymbol{I}_d\right), a \sim N(0, 1)$ and given $\boldsymbol{x}_i, \boldsymbol{x}_j \in \Omega$. Then we have*

*(i) if* $\mathrm{X} := \sigma\left(\boldsymbol{w}^\top \boldsymbol{x}_i\right) \sigma\left(\boldsymbol{x} \cdot \boldsymbol{x}_j\right)$, *then* $\|\mathrm{X}\|_{\psi_1} \leq cdC_{\psi,d}$, *where c is the constant shown in Assumption A.1.*

*(ii) if* $\mathrm{X} := a^2 \sigma'\left(\boldsymbol{w}^\top \boldsymbol{x}_i\right) \sigma'\left(\boldsymbol{w}^\top \boldsymbol{x}_j\right) \boldsymbol{x}_i \cdot \boldsymbol{x}_j$, *then* $\|\mathrm{X}\|_{\psi_1} \leq cdC_{\psi,d}$.

*Proof.* (i) $|\mathrm{X}| \leq d\|\boldsymbol{w}\|_2^2 = d\mathrm{Z}$ and

$$
\begin{aligned}
\|\mathrm{X}\|_{\psi_1} &= \inf\left\{s > 0 \mid \mathbf{E}_\mathrm{X} \exp(|\mathrm{X}|/s) \leq 2\right\} \\
&= \inf\left\{s > 0 \mid \mathbf{E}_{\boldsymbol{w}} \exp\left(\left|\sigma\left(\boldsymbol{w}^\top \boldsymbol{x}_i\right) \sigma\left(\boldsymbol{w}^\top \boldsymbol{x}_j\right)\right|/s\right) \leq 2\right\} \\
&\leq \inf\left\{s > 0 \mid \mathbf{E}_{\boldsymbol{w}} \exp\left(cd\|\boldsymbol{w}\|_2^2/s\right) \leq 2\right\} \\
&= \inf\left\{s > 0 \mid \mathbf{E}_\mathrm{Z} \exp(cd|\mathrm{Z}|/s) \leq 2\right\} \\
&= cd \inf\left\{s > 0 \mid \mathbf{E}_\mathrm{Z} \exp(|\mathrm{Z}|/s) \leq 2\right\} \\
&= cd\left\|\chi^2(d)\right\|_{\psi_1} \\
&\leq cdC_{\psi,d}
\end{aligned}
$$

(ii) $|\mathrm{X}| \leq cd|a|^2 \leq cd\mathrm{Z}$ and $\|\mathrm{X}\|_{\psi_1} \leq cdC_{\psi,d}$. $\qquad\square$

**Proposition A.5** (sub-exponential Bernstein's inequality (Vershynin, 2018)). *Suppose that $\mathrm{X}_1, \ldots, \mathrm{X}_m$ are i.i.d. sub-exponential random variables with $\mathbf{E}\mathrm{X}_1 = \mu$, then for any $s \geq 0$ we have*

$$
\mathbf{P}\left(\left|\frac{1}{m}\sum_{k=1}^m \mathrm{X}_k - \mu\right| \geq s\right) \leq 2\exp\left(-C_0 m \min\left(\frac{s^2}{\|\mathrm{X}_1\|_{\psi_1}^2}, \frac{s}{\|\mathrm{X}_1\|_{\psi_1}}\right)\right),
$$

*where $C_0$ is an absolute constant.*

**Proposition A.6.** *Suppose that Assumption A.1 holds, and given $\delta \in (0, 1)$, $\boldsymbol{w} \sim N\left(0, \boldsymbol{I}_d\right), a \sim N(0, 1)$ and the sample set $S = \{\boldsymbol{x}_i\}_{i=1}^n \subset \Omega$ with $\boldsymbol{x}_i$ 's drawn i.i.d. with uniformly distributed with and any $i, j \in [n], i \neq j, \boldsymbol{x}_i \nparallel \boldsymbol{x}_j$. If $m \geq \frac{16n^2 c^2 d^2 C_{\psi,d}}{C_0 \lambda^2} \log \frac{4n^2}{\delta}$ then with probability at least $1 - \delta$ over the choice of $\boldsymbol{\theta}(0)$, we have*

$$
\lambda_{\min}\left(\boldsymbol{S}\boldsymbol{S}^\top\right) \geq \frac{3}{4}(\lambda_{min}(\boldsymbol{K}^{[a]}) + \lambda_{min}(\boldsymbol{K}^{[\boldsymbol{\omega}]})).
$$

*Proof.* Recall that

$$
\boldsymbol{S} = \left[\nabla_{\boldsymbol{\theta}} u_{\boldsymbol{\theta}}(\boldsymbol{x}_1), \ldots, \nabla_{\boldsymbol{\theta}} u_{\boldsymbol{\theta}}(\boldsymbol{x}_n)\right], \tag{36}
$$

and $\boldsymbol{\theta}$ contain two parts, $a$ and $\boldsymbol{w}$ parts, therefore $\boldsymbol{S}\boldsymbol{S}^\top$ can be rewrite as $\boldsymbol{S}_a \boldsymbol{S}_a^\top + \boldsymbol{S}_{\boldsymbol{w}} \boldsymbol{S}_{\boldsymbol{w}}^\top$ where

$$
\boldsymbol{S}_a = \left[\nabla_a u_{\boldsymbol{\theta}}(\boldsymbol{x}_1), \ldots, \nabla_a u_{\boldsymbol{\theta}}(\boldsymbol{x}_n)\right], \; \boldsymbol{S}_{\boldsymbol{w}} = \left[\nabla_{\boldsymbol{w}} u_{\boldsymbol{\theta}}(\boldsymbol{x}_1), \ldots, \nabla_{\boldsymbol{w}} u_{\boldsymbol{\theta}}(\boldsymbol{x}_n)\right] \tag{37}
$$

For any $\varepsilon > 0$, we define

$$
\Omega_{ij}^{[a]} := \left\{\boldsymbol{\theta} \mid \left|(\boldsymbol{S}_a \boldsymbol{S}_a^\top)_{ij}(\boldsymbol{\theta}) - K_{ij}^{[a]}\right| \leq \frac{\varepsilon}{n}\right\}, \; \Omega_{ij}^{[\boldsymbol{w}]} := \left\{\boldsymbol{\theta} \mid \left|(\boldsymbol{S}_{\boldsymbol{w}} \boldsymbol{S}_{\boldsymbol{w}}^\top)_{ij}(\boldsymbol{\theta}) - K_{ij}^{[\boldsymbol{w}]}\right| \leq \frac{\varepsilon}{n}\right\}. \tag{38}
$$

Setting $\varepsilon \leq cndC_{\psi,d}$, by Proposition A.5 and Lemma A.4, we have

$$
\mathbf{P}(\Omega_{ij}^{[a]}) \geq 1 - 2\exp\left(-\frac{mC_0\varepsilon^2}{n^2 d^2 c^2 C_{\psi,d}}\right), \; \mathbf{P}(\Omega_{ij}^{[\boldsymbol{w}]}) \geq 1 - 2\exp\left(-\frac{mC_0\varepsilon^2}{n^2 d^2 c^2 C_{\psi,d}}\right). \tag{39}
$$

Due to inclusion-exclusion Principle, we have

$$
\mathbf{P}\left(\left\{\boldsymbol{\theta} \mid \left\|\boldsymbol{S}_a \boldsymbol{S}_a^\top(\boldsymbol{\theta}) - \boldsymbol{K}^{[a]}\right\|_F \leq \varepsilon\right\} \cap \left\{\boldsymbol{\theta} \mid \left\|\boldsymbol{S}_{\boldsymbol{w}} \boldsymbol{S}_{\boldsymbol{w}}^\top(\boldsymbol{\theta}) - \boldsymbol{K}^{[\boldsymbol{w}]}\right\|_F \leq \varepsilon\right\}\right) \geq \sum_{i,j=1}^n \left(\mathbf{P}(\Omega_{ij}^{[a]}) + \mathbf{P}(\Omega_{ij}^{[\boldsymbol{w}]})\right) - 2n^2 - 1,
$$

$$
\tag{40}
$$

therefore, with probability at least

$$1 - 4n^2 \exp\left(-\frac{mC_0\varepsilon^2}{n^2d^2c^2C_{\psi,d}^2}\right)$$

over the choice of $\boldsymbol{\theta}$, we have

$$\left\|\boldsymbol{S}_a\boldsymbol{S}_a^\top(\boldsymbol{\theta}) - \boldsymbol{K}^{[a]}\right\|_F \le \varepsilon, \ \left\|\boldsymbol{S}_{\boldsymbol{w}}\boldsymbol{S}_{\boldsymbol{w}}^\top(\boldsymbol{\theta}) - \boldsymbol{K}^{[\boldsymbol{w}]}\right\|_F \le \varepsilon \tag{41}$$

Hence by taking $\varepsilon = \frac{\lambda_1}{4}$ and $\delta = 4n^2\exp\left(-\frac{mC_0\lambda_1^2}{16n^2d^2c^2C_{\psi,d}^2}\right)$, where $\lambda_1 = \min\{\lambda_{\min}(\boldsymbol{K}^{[a]}), \lambda_{\min}(\boldsymbol{K}^{[\boldsymbol{\omega}]})\}$

$$\begin{aligned}
\lambda_{\min}\left(\boldsymbol{S}\boldsymbol{S}^\top\right) &\ge \lambda_{\min}\left(\boldsymbol{S}_a\boldsymbol{S}_a^\top\right) + \lambda_{\min}\left(\boldsymbol{S}_{\boldsymbol{\omega}}\boldsymbol{S}_{\boldsymbol{\omega}}^\top\right) \\
&\ge \lambda_{\min}(\boldsymbol{K}^{[a]}) + \lambda_{\min}(\boldsymbol{K}^{[\boldsymbol{\omega}]}) - \left\|\boldsymbol{S}_a\boldsymbol{S}_a^\top(\boldsymbol{\theta}) - \boldsymbol{K}^{[a]}\right\|_F - \left\|\boldsymbol{S}_{\boldsymbol{w}}\boldsymbol{S}_{\boldsymbol{w}}^\top(\boldsymbol{\theta}) - \boldsymbol{K}^{[\boldsymbol{w}]}\right\|_F \\
&\ge \frac{3}{4}(\lambda_{\min}(\boldsymbol{K}^{[a]}) + \lambda_{\min}(\boldsymbol{K}^{[\boldsymbol{\omega}]})).
\end{aligned} \tag{42}$$

$\square$

Combining Lemma A.2 and Proposition A.6, we obtain that under the conditions stated in Proposition A.6, the following holds with high probability:

$$\lambda_{\min}(\boldsymbol{S}\boldsymbol{S}^\top) > 0. \tag{43}$$

### A.2. Proof of Theorem 4.1

We can analysis the smallest eigenvalue of the problems based on the following lemma:

**Lemma A.7** ((Li & Mathias, 1999)). *Let $\boldsymbol{A}$ be an $n \times n$ Hermitian matrix and let $\tilde{\boldsymbol{A}} = \boldsymbol{T}^*\boldsymbol{A}\boldsymbol{T}$. Then we have*

$$\lambda_{min}\left(\boldsymbol{T}^*\boldsymbol{T}\right) \le \frac{\lambda_{min}(\tilde{\boldsymbol{A}})}{\lambda_{min}(\boldsymbol{A})} \le \lambda_{max}\left(\boldsymbol{T}^*\boldsymbol{T}\right).$$

*Proof of Theorem 4.1.* We first show that $\lambda_{\min}(\boldsymbol{K}_\varepsilon) > 0$, which follows directly from Lemma A.7:

$$\lambda_{\min}(\boldsymbol{K}_\varepsilon) \ge \lambda_{\min}(\boldsymbol{S}\boldsymbol{S}^\top) \cdot \lambda_{\min}(\boldsymbol{D}_\varepsilon\boldsymbol{D}_\varepsilon^\top) > 0.$$

Therefore, at the beginning of gradient descent, the kernel of the gradient descent step is strictly positive. We then define $T$ as

$$T := \inf\{t \mid \boldsymbol{\theta}(t) \notin N(\boldsymbol{\theta}(0))\}, \tag{44}$$

where

$$N(\boldsymbol{\theta}) := \left\{\boldsymbol{\theta} \mid \|\boldsymbol{K}_\varepsilon(\boldsymbol{\theta}(t)) - \boldsymbol{K}_\varepsilon(\boldsymbol{\theta}(0))\|_F \le \frac{1}{2}\lambda_{\min}(\boldsymbol{K}_\varepsilon)\right\}.$$

We now analyze the evolution of the loss function:

$$\begin{aligned}
\frac{\mathrm{d}L_H(\boldsymbol{\theta}(t))}{\mathrm{d}t} &= \nabla_{\boldsymbol{\theta}}L_H(\boldsymbol{\theta})\frac{\mathrm{d}\boldsymbol{\theta}}{\mathrm{d}t} \\
&= -\frac{1}{n^2}\boldsymbol{l}_\varepsilon\boldsymbol{D}_\varepsilon\boldsymbol{S}\boldsymbol{S}^\top\boldsymbol{D}_\varepsilon^\top\boldsymbol{l}_\varepsilon^\top \\
&\le -\frac{2}{n}\lambda_{\min}(\boldsymbol{K}_\varepsilon(\boldsymbol{\theta}(t)))L(\boldsymbol{\theta}(t)),
\end{aligned} \tag{45}$$

where we use the fact that $\boldsymbol{l}_\varepsilon \cdot \boldsymbol{l}_\varepsilon^\top = 2nL_H(\boldsymbol{\theta}(t))$.

Furthermore, for $t \in [0, T]$, we have

$$\|\boldsymbol{K}_\varepsilon(\boldsymbol{\theta}(t)) - \boldsymbol{K}_\varepsilon(\boldsymbol{\theta}(0))\|_F \le \frac{1}{2}\lambda_{\min}(\boldsymbol{K}_\varepsilon).$$

This implies

$$
\begin{aligned}
\lambda_{\min}(\boldsymbol{K}_\varepsilon(\boldsymbol{\theta}(t))) &\geq \lambda_{\min}\Big(\boldsymbol{K}_\varepsilon(\boldsymbol{\theta}(t)) - \boldsymbol{K}_\varepsilon(\boldsymbol{\theta}(0))\Big) + \lambda_{\min}\Big(\boldsymbol{K}_\varepsilon(\boldsymbol{\theta}(0))\Big) \\
&\geq \lambda_{\min}\Big(\boldsymbol{K}_\varepsilon(\boldsymbol{\theta}(0))\Big) - \sigma_{\min}\Big(\boldsymbol{K}_\varepsilon(\boldsymbol{\theta}(t)) - \boldsymbol{K}_\varepsilon(\boldsymbol{\theta}(0))\Big) \\
&\geq \lambda_{\min}\Big(\boldsymbol{K}_\varepsilon(\boldsymbol{\theta}(0))\Big) - \|\boldsymbol{K}_\varepsilon(\boldsymbol{\theta}(t)) - \boldsymbol{K}_\varepsilon(\boldsymbol{\theta}(0))\|_F \\
&\geq \frac{1}{2}\lambda_{\min}\Big(\boldsymbol{K}_\varepsilon(\boldsymbol{\theta}(0))\Big).
\end{aligned}
$$

Therefore, we obtain

$$
\frac{\mathrm{d}L_H(\boldsymbol{\theta}(t))}{\mathrm{d}t} \leq -\frac{1}{n}\lambda_{\min}(\boldsymbol{K}_\varepsilon(\boldsymbol{\theta}(0)))L_H(\boldsymbol{\theta}(t)), \tag{46}
$$

for $t \in [0, T]$. Solving this differential inequality yields

$$
L_H(\boldsymbol{\theta}(t)) \leq L_H(\boldsymbol{\theta}(0)) \exp\left(-\frac{\lambda_{\min}(\boldsymbol{K}_\varepsilon)}{n}t\right) \tag{47}
$$

for all $t \in [0, T]$.

Finally, for the inequality

$$
\lambda_{\min}(\boldsymbol{S}\boldsymbol{S}^\top)\lambda_{\min}(\boldsymbol{D}_\varepsilon\boldsymbol{D}_\varepsilon^\top) \leq \lambda_{\min}(\boldsymbol{K}_\varepsilon) \leq \lambda_{\min}(\boldsymbol{S}\boldsymbol{S}^\top)\lambda_{\max}(\boldsymbol{D}_\varepsilon\boldsymbol{D}_\varepsilon^\top), \tag{48}
$$

it follows directly from Lemma A.7. □

### A.3. Proof of Theorem 4.3

*Proof of Theorem 4.3.* First, we have

$$
\begin{aligned}
u(\varepsilon_{k+1}) &= u(\varepsilon_k) + (\varepsilon_{k+1} - \varepsilon_k)u'(\varepsilon_k) + \frac{1}{2}(\varepsilon_{k+1} - \varepsilon_k)^2 u''(\xi_k) \\
&= u(\varepsilon_k) + (\varepsilon_{k+1} - \varepsilon_k)h(\varepsilon_k, u(\varepsilon_k)) + \frac{1}{2}(\varepsilon_{k+1} - \varepsilon_k)^2 u''(\xi_k),
\end{aligned} \tag{49}
$$

where $\xi_k$ lies between $\varepsilon_{k+1}$ and $\varepsilon_k$ and depends on $\boldsymbol{x}$. Therefore, we obtain

$$
e(\varepsilon_{k+1}) = e(\varepsilon_k) + (\varepsilon_{k+1} - \varepsilon_k)(h(\varepsilon_k, u(\varepsilon_k)) - h(\varepsilon_k, U(\varepsilon_k))) + \frac{1}{2}(\varepsilon_{k+1} - \varepsilon_k)^2 u''(\xi_k), \tag{50}
$$

where $e(\varepsilon_k) = u(\varepsilon_k) - U(\varepsilon_k)$. Then, we have

$$
\begin{aligned}
&\|e(\varepsilon_{k+1})\|_{H^2(\Omega)} \\
={}&\|e(\varepsilon_k)\|_{H^2(\Omega)} + (\varepsilon_{k+1} - \varepsilon_k)\|h(\varepsilon_k, u(\varepsilon_k)) - h(\varepsilon_k, U(\varepsilon_k))\|_{H^2(\Omega)} \\
&+ \frac{1}{2}(\varepsilon_{k+1} - \varepsilon_k)^2\|u''(\xi_k)\|_{H^2(\Omega)} \\
\leq{}&\|e(\varepsilon_k)\|_{H^2(\Omega)} + (\varepsilon_{k+1} - \varepsilon_k)P_{\varepsilon_k}\|e(\varepsilon_k)\|_{H^2(\Omega)} + \frac{1}{2}\frac{\varepsilon_0 - \varepsilon_n}{n}\tau \\
\leq{}&\|e(\varepsilon_k)\|_{H^2(\Omega)} + P \cdot \frac{\varepsilon_0 - \varepsilon_n}{n}\|e(\varepsilon_k)\|_{H^2(\Omega)} + \frac{1}{2}\frac{\varepsilon_0 - \varepsilon_n}{n}\tau.
\end{aligned} \tag{51}
$$

Recalling that $e_0 = \|u(\varepsilon_0) - U(\varepsilon_0)\|_{H^2(\Omega)}$, we obtain

$$
\begin{aligned}
\|e(\varepsilon_n)\|_{H^2(\Omega)} &\leq e_0\left(1 + P \cdot \frac{\varepsilon_0 - \varepsilon_n}{n}\right)^n + \frac{\tau}{2}\frac{\varepsilon_0 - \varepsilon_n}{n}\sum_{n=0}^{n-1}\left(1 + P \cdot \frac{\varepsilon_0 - \varepsilon_n}{n}\right)^n \\
&= e_0\left(1 + P \cdot \frac{\varepsilon_0 - \varepsilon_n}{n}\right)^n + \frac{\tau}{2}\frac{\left(1 + P \cdot \frac{\varepsilon_0 - \varepsilon_n}{n}\right)^n - 1}{P} \\
&\leq \frac{\tau(e^{P(\varepsilon_0 - \varepsilon_n)} - 1)}{2P} + e_0 e^{P(\varepsilon_0 - \varepsilon_n)},
\end{aligned} \tag{52}
$$

where the last step follows from the inequality

$$(1 + a)^m \leq e^{ma},$$

for $a > 0$.  □

**Corollary A.8** (Convergence of Homotopy Functions). *Suppose the assumptions in Theorem 4.3 hold, and $H(\varepsilon_n, u)$ is Lipschitz continuous in $H^2(\Omega)$, i.e.,*

$$\|H(u_1, \varepsilon_n) - H(u_2, \varepsilon_n)\|_{H^2(\Omega)} \leq L\|u_1 - u_2\|_{H^2(\Omega)}.$$

*Then, we have*

$$
\begin{aligned}
&\|H(U(\varepsilon_n), \varepsilon_n)\|_{H^2(\Omega)} \\
&\leq L\left[e_0 e^{P(\varepsilon_0 - \varepsilon_n)} + \frac{\tau(e^{P(\varepsilon_0 - \varepsilon_n)} - 1)}{2P}\right] \ll 1.
\end{aligned}
\tag{53}
$$

*Proof.* The proof follows directly from the result in Theorem 4.3.  □

### A.4. Discussion on $e_0$

In Theorem 4.3 and Corollary A.8, one important assumption is that we assume $e_0$ is small. Here, we discuss why this assumption is reasonable.

First, we use physics-informed neural networks (PINNs) to solve the following equations:

$$
\begin{cases}
\mathcal{L}_\varepsilon u = f(u), & \text{in } \Omega, \\
\mathcal{B}u = g(x), & \text{on } \partial\Omega,
\end{cases}
\tag{54}
$$

where $\mathcal{L}_\varepsilon$ is a differential operator defining the PDE with certain parameters, $\mathcal{B}$ is an operator associated with the boundary and/or initial conditions, and $\Omega \subseteq \mathbb{R}^d$.

The corresponding continuum loss function is given by:

$$
L_c(\boldsymbol{\theta}) := \frac{1}{2}\int_\Omega (\mathcal{L}_\varepsilon u(\boldsymbol{x}; \boldsymbol{\theta}) - f(u))^2 \, \mathrm{d}\boldsymbol{x} + \frac{\lambda}{2}\int_{\partial\Omega} (\mathcal{B}u(\boldsymbol{x}; \boldsymbol{\theta}) - g(\boldsymbol{x}))^2 \, \mathrm{d}\boldsymbol{x}.
\tag{55}
$$

We assume this loss function satisfies a regularity condition:

**Assumption A.9.** Let $u_*$ be the exact solution of Eq. (54). Then, there exists a constant $C$ such that

$$
\|u(\boldsymbol{x}; \boldsymbol{\theta}) - u_*(\boldsymbol{x})\|_{H^2(\Omega)} \leq CL_c(\boldsymbol{\theta}).
\tag{56}
$$

The above assumption holds in many cases. For example, based on (Grisvard, 2011), when $\mathcal{L}$ is a linear elliptic operator with smooth coefficients, and $f(u)$ reduces to $f(\boldsymbol{x}) \in L^2(\Omega)$, and if $\Omega$ is a polygonal domain (e.g., $[0,1]^d$), then, provided the boundary conditions are always satisfied, the assumption holds.

Therefore, we only need to ensure that $L_c(\boldsymbol{\theta}_s)$ is sufficiently small, where $\boldsymbol{\theta}_s$ denotes the learned parameters at convergence. Here, $L_c(\boldsymbol{\theta}_s)$ can be divided into three sources of error: approximation error, generalization error, and training error:

$$
\begin{aligned}
\boldsymbol{\theta}_c &= \arg\min_{\boldsymbol{\theta}} L_c(\boldsymbol{\theta}) = \arg\min_{\boldsymbol{\theta}} \frac{1}{2}\int_\Omega (\mathcal{L}_\varepsilon u(\boldsymbol{x}; \boldsymbol{\theta}) - f(u(\boldsymbol{x})))^2 \, \mathrm{d}\boldsymbol{x} + \frac{\lambda}{2}\int_{\partial\Omega}(\mathcal{B}u(\boldsymbol{x}; \boldsymbol{\theta}) - g(\boldsymbol{x}))^2 \, \mathrm{d}\boldsymbol{x}, \\
\boldsymbol{\theta}_d &= \arg\min_{\boldsymbol{\theta}} L(\boldsymbol{\theta}) = \arg\min_{\boldsymbol{\theta}} \frac{1}{2n_r}\sum_{i=1}^{n_r} \left(\mathcal{L}_\varepsilon u(\boldsymbol{x}_r^i; \boldsymbol{\theta}) - f(u(\boldsymbol{x}_r^i; \boldsymbol{\theta}))\right)^2 + \frac{\lambda}{2n_b}\sum_{j=1}^{n_b}\left(\mathcal{B}u(\boldsymbol{x}_b^j; \boldsymbol{\theta}) - g(\boldsymbol{x}_b^j)\right)^2,
\end{aligned}
\tag{57}
$$

where $\boldsymbol{x}_r^i, \boldsymbol{x}_b^j$ are sampled points as defined in Eq. (2).

The error decomposition can then be expressed as:

$$\mathbb{E}L_c(\boldsymbol{\theta}_s) \leq L_c(\boldsymbol{\theta}_c) + \mathbb{E}L(\boldsymbol{\theta}_c) - L_c(\boldsymbol{\theta}_c) + \mathbb{E}L(\boldsymbol{\theta}_d) - \mathbb{E}L(\boldsymbol{\theta}_c) + \mathbb{E}L(\boldsymbol{\theta}_s) - \mathbb{E}L(\boldsymbol{\theta}_d) + \mathbb{E}L_c(\boldsymbol{\theta}_s) - \mathbb{E}L(\boldsymbol{\theta}_s)$$

$$\leq \underbrace{L_c(\boldsymbol{\theta}_c)}_{\text{approximation error}} + \underbrace{\mathbb{E}L(\boldsymbol{\theta}_c) - L_c(\boldsymbol{\theta}_c) + \mathbb{E}L_c(\boldsymbol{\theta}_s) - \mathbb{E}L(\boldsymbol{\theta}_s)}_{\text{generalization error}} + \underbrace{\mathbb{E}L(\boldsymbol{\theta}_s) - \mathbb{E}L(\boldsymbol{\theta}_d)}_{\text{training error}}, \tag{58}$$

where the last inequality is due to $\mathbb{E}L(\boldsymbol{\theta}_d) - \mathbb{E}L(\boldsymbol{\theta}_c) \leq 0$ based on the definition of $\boldsymbol{\theta}_d$.

The approximation error describes how closely the neural network approximates the exact solution of the PDEs. If $f$ is a Lipschitz continuous function, $\mathcal{L}_\varepsilon$ is Lipschitz continuous from $W^{2,1}(\Omega) \to L^1(\Omega)$, and $\mathcal{B}$ is Lipschitz continuous from $L^1(\partial\Omega) \to L^1(\partial\Omega)$, with $u(\boldsymbol{x};\boldsymbol{\theta}), u_* \in W^{2,\infty}(\bar{\Omega})$ and $\partial\Omega \in C^1(\Omega)$, then we have

$$L_c(\boldsymbol{\theta}) = \int_\Omega \left(\mathcal{L}_\varepsilon u(\boldsymbol{x};\boldsymbol{\theta}) - f(u(\boldsymbol{x}))\right)^2 - \left(\mathcal{L}_\varepsilon u_* - f(u_*)\right)^2 \, d\boldsymbol{x} + \frac{\lambda}{2}\int_{\partial\Omega}\left(\mathcal{B}u(\boldsymbol{x};\boldsymbol{\theta}) - g(\boldsymbol{x})\right)^2 - \left(\mathcal{B}u_* - g(\boldsymbol{x})\right)^2 \, d\boldsymbol{x}$$

$$\leq C_1\left(\|\mathcal{L}_\varepsilon(u(\boldsymbol{x};\boldsymbol{\theta}) - u_*)\|_{L^1(\Omega)} + \|f(u(\boldsymbol{x};\boldsymbol{\theta})) - f(u_*)\|_{L^1(\Omega)}\right) + C_2\|\mathcal{B}(u(\boldsymbol{x};\boldsymbol{\theta}) - u_*)\|_{L^1(\partial\Omega)}$$

$$\leq C_3\|u(\boldsymbol{x};\boldsymbol{\theta}) - u_*\|_{W^{2,1}(\Omega)} + C_4\|u(\boldsymbol{x};\boldsymbol{\theta}) - u_*\|_{W^{1,1}(\Omega)}$$

$$\leq C\|u(\boldsymbol{x};\boldsymbol{\theta}) - u_*\|_{W^{2,1}(\Omega)}, \tag{59}$$

where the second inequality follows from the trace theorem (Evans, 2022). Therefore, we conclude that $L_c(\boldsymbol{\theta})$ can be bounded by $\|u(\boldsymbol{x};\boldsymbol{\theta}) - u_*\|_{W^{2,1}(\Omega)}$, which has been widely studied in the context of shallow neural networks (Siegel & Xu, 2020) and deep neural networks (Yang et al., 2023). These results show that if the number of neurons is sufficiently large, the error in this part becomes small.

For the generalization error, it arises from the fact that we have only a finite number of data points. This error can be bounded using Rademacher complexity (Yang et al., 2023; Luo & Yang, 2020), which leads to a bound of $\mathcal{O}\left(n_r^{-\frac{1}{2}}\right) + \mathcal{O}\left(n_b^{-\frac{1}{2}}\right)$. In other words, this error term is small when the number of sample points is large.

For the training error, Theorem 4.1 shows that when $\varepsilon$ is large in certain PDEs, the loss function can decay efficiently, reducing the training error to a small value.

## B. Details on Experiments

### B.1. Overall Experiments Settings

**Examples.** We conduct experiments on function learning case: 1D Allen-Cahn equation, 2D Allen-Cahn equation, high dimension Helmholz equation, high frequency function approximation and operator learning for Burgers' equation. These equations have been studied in previous works investigating difficulties in solving numerically; we use the formulations in Xu (2020); Zhang et al. (2024); Hao & Yang (2019) for our experiments.

**Network Structure.** We use multilayer perceptrons (MLPs) with tanh activations and three hidden layers with width 30. We initialize these networks with the Xavier normal initialization (Glorot & Bengio, 2010) and all biases equal to zero.

**Training.** We use Adam to train the neural network and we tune the learning rate by a grid search on $\{10^{-5}, 10^{-4}, 10^{-3}, 10^{-2}\}$. All iterations continue until the loss stabilizes and no longer decreases significantly.

**Device.** We develop our experiments in PyTorch 1.12.1 (Paszke et al., 2019) with Python 3.9.12. Each experiment is run on a single NVIDIA 3070Ti GPU using CUDA 11.8. As summarized in Table 6, we report the training cost (per epoch and total epochs) for experiments in our paper.

### B.2. 1D Allen-Cahn Equation

Number of residual points $n_{\text{res}} = 200$ and number of boundary points $n_{\text{bc}} = 2$. In this example, we use forward Euler method to numerically solve the homotopy dynamics. And $\varepsilon_0 = 0.1$ and $\varepsilon_n = 0.01$, here we choose $\Delta\varepsilon_k = 0.001$. Here, we use strategy 1 to train the neural network.

The results for using original training for this example Figure 7. As shown in the figure, the original training method results in a large training error, leading to poor accuracy.

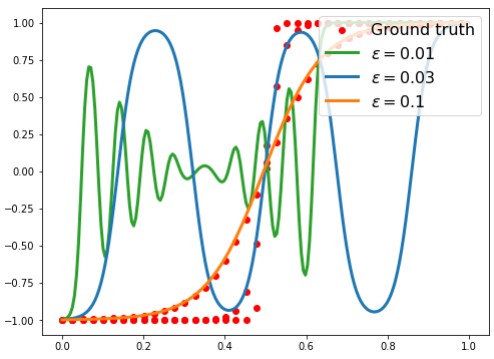

*Figure 7.* Solution for 1D Allen-Cahn equation for origin training.

*Table 6.* **Training Cost for Different Examples.** The table summarizes training time per epoch and total number of epochs for various PDE problems.

| Example | 1D Allen-Cahn Equation | Example 5.1 | Example 5.2 | Example 5.3 |
|---|---|---|---|---|
| Training Time per Epoch | 0.05s | 0.09s | 0.01s | 0.4s |
| Total Epochs (Steps) | $1.0 \times 10^3$ | $4.0 \times 10^6$ | $4.0 \times 10^6$ | $2.0 \times 10^6$ |

### B.3. 2D Allen-Cahn Equation

Number of residual points $n_{res} = 50 \times 50$ and number of boundary points $n_{bc} = 198$. For a fair comparison, all methods were implemented using the same neural network architecture, specifically a fully connected network with layer sizes $[2, 30, 30, 30, 1]$. In this example, we optimize using the Homotopy Loss. We set $s_0 = 1.0$ and $s_n = 0$, initially choosing $\Delta s = 0.1$, and later refining it to $\Delta t = 0.01$. When $s = 0.05, \varepsilon(s) = 0.05$ we fix $\varepsilon = 0.05$ and gradually decrease $s$ to 0.

The reference ground truth solution is obtained using the finite difference method with $N = 1000 \times 1000$ grid points. The result is shown in Figure 8.

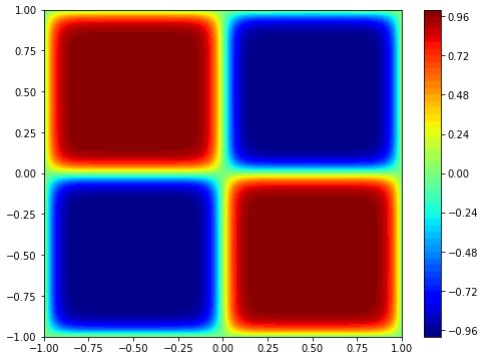

*Figure 8.* Reference Solution for 2D Allen-Cahn equation.

The result obtained using PINN is shown in the Figure 9. It is evident that the solution still deviates significantly from the ground truth solution.

The result obtained using curriculum regularity strategy (Krishnapriyan et al., 2021) is shown in the Figure 10 below, where $\Delta\varepsilon = 0.01$.

The result obtained using Sequence-to-sequence training strategy (Wight & Zhao, 2020; Mattey & Ghosh, 2022) is shown in the Figure 11 below, where $\Delta t = 0.01$.

The result obtained using resampling strategy is shown in the Figure 12. In all resampling strategies, additional sample points are eventually concentrated near the sharp interface region. In our comparative experiments, we start with a uniform

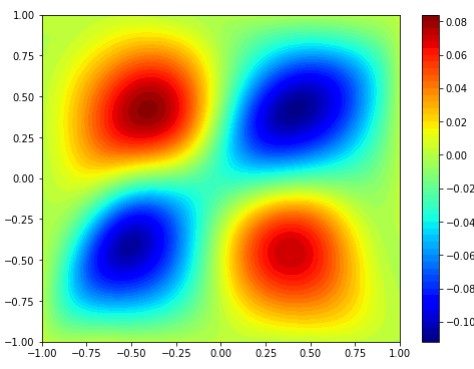

*Figure 9.* Solution for 2D Allen-Cahn equation for origin training.

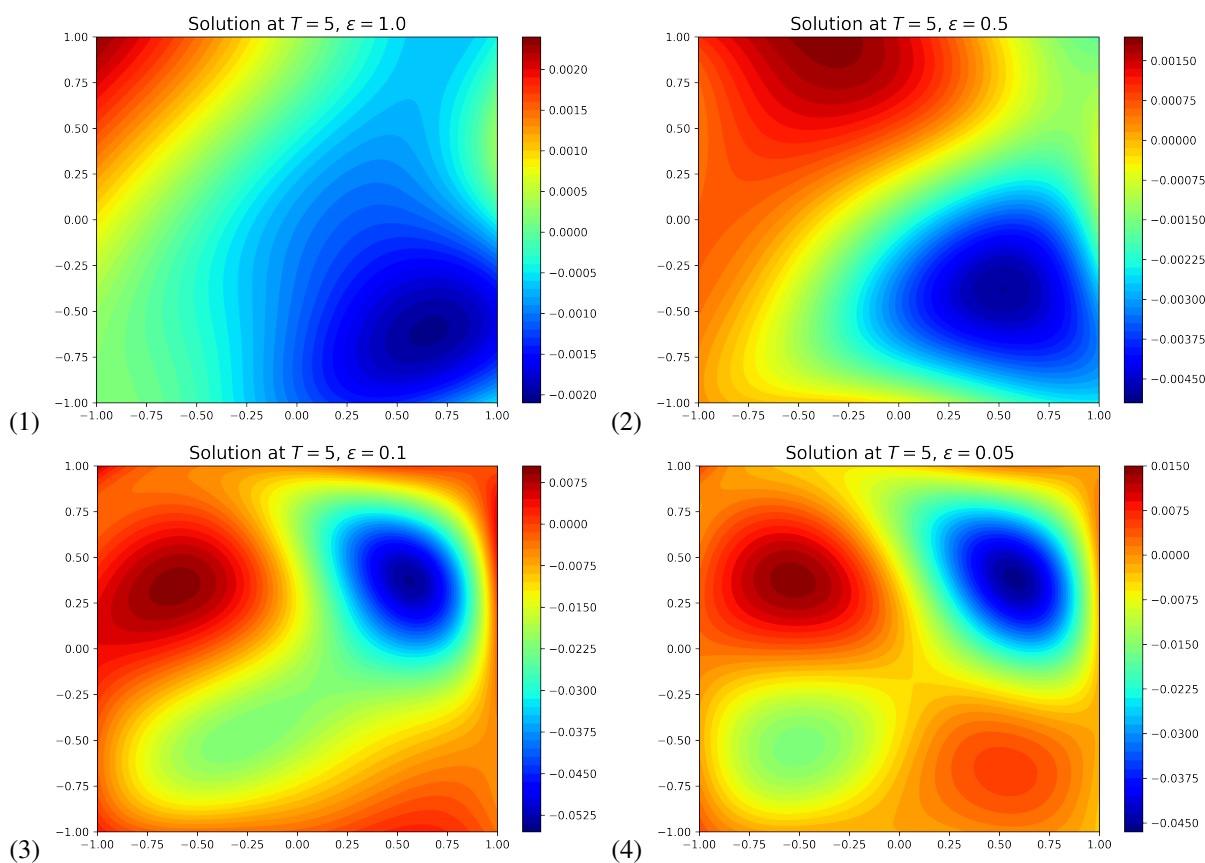

*Figure 10.* Numerical results for the 2D Allen–Cahn equation using the Curriculum training strategy.

grid of $50 \times 50$ sample points and augment it by adding $5000$ points in the vicinity of the sharp interface.

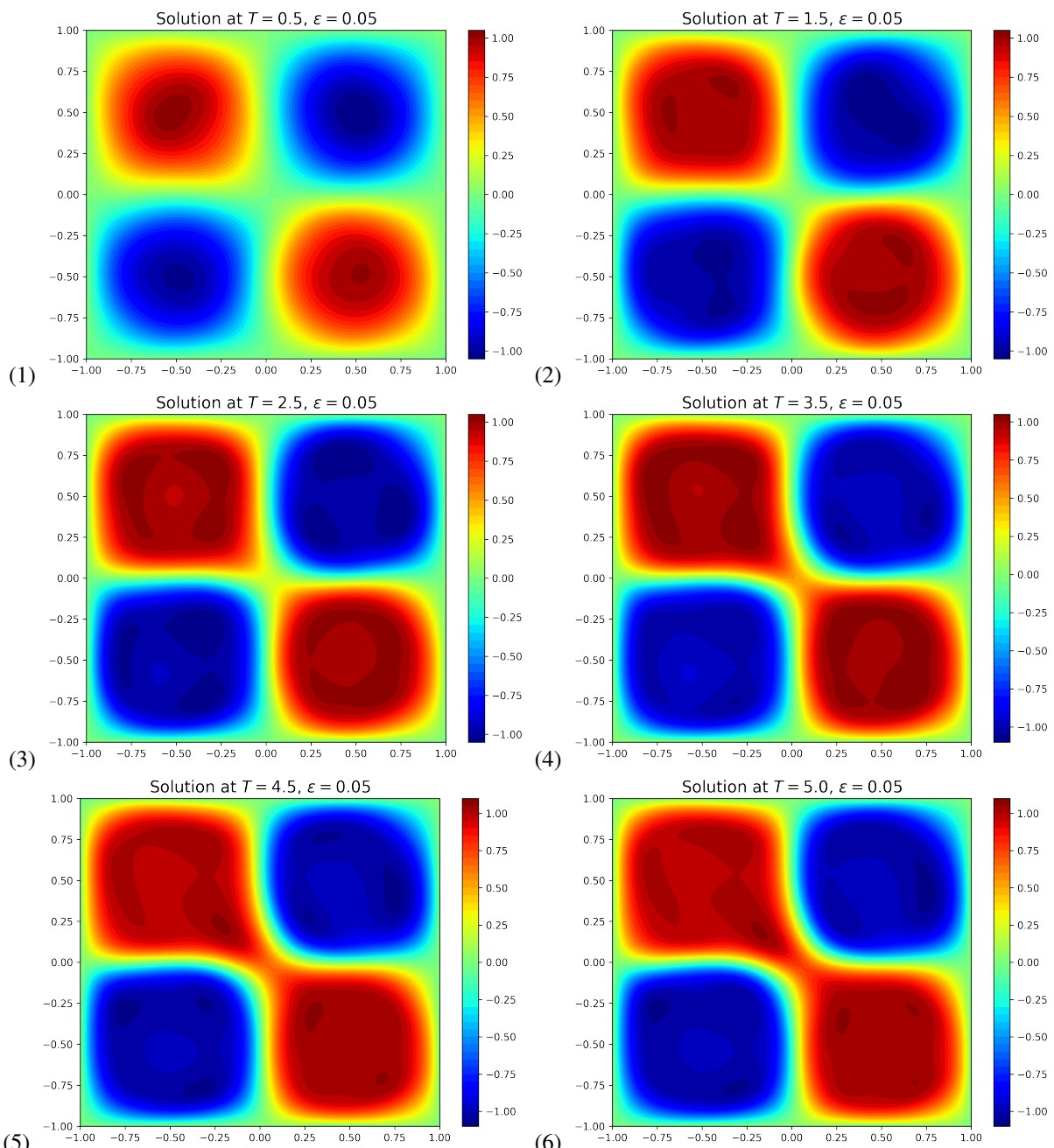

*Figure 11.* Numerical results for the 2D Allen–Cahn equation using the Sequence-to-sequence training strategy.

## B.4. High Dimension Helmholtz Equation

Number of residual points $n_{\text{res}} = 10000$ and number of boundary points $n_{\text{bc}} = 2000$. Neural network architecture is a fully connected network with layer sizes $[2, 30, 30, 30, 1]$. In this example, we optimize using the Homotopy Loss. We set $\varepsilon_0 = 1.0$, initially choosing $\Delta\varepsilon = 0.1$, and later refining it to $\Delta\varepsilon = 0.01$ until $\varepsilon_n = \frac{1}{50}$.

**Largest eigenvalue of $D_\varepsilon$.** As shown in Figure 13, a smaller $\varepsilon$ results in a smaller largest eigenvalue of (60), leading to a slower convergence rate and increased difficulty in training.

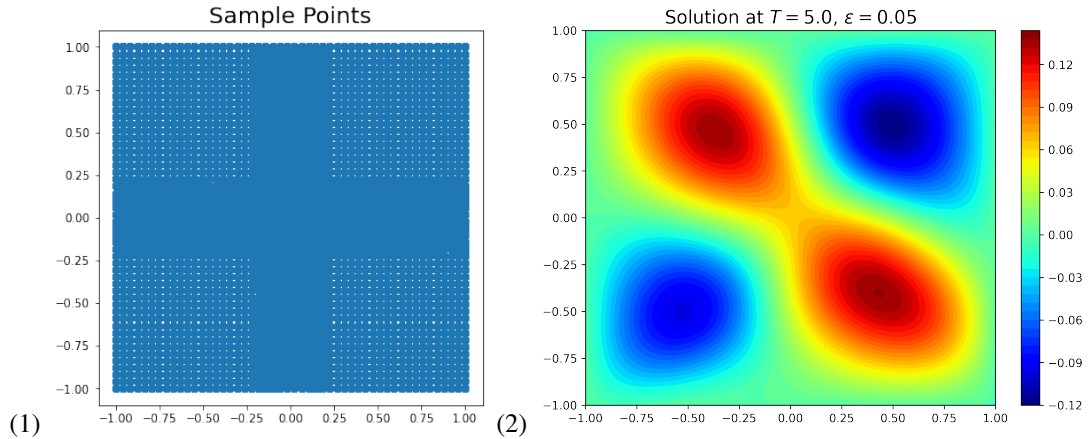

*Figure 12.* Numerical results for the 2D Allen–Cahn equation using the Resampling training strategy.

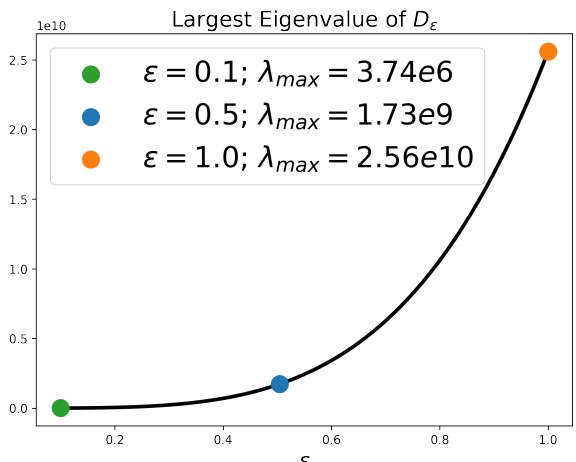

*Figure 13.* Largest eigenvalue of $\boldsymbol{D}_\varepsilon$ (60) for different $\varepsilon$. A smaller $\varepsilon$ results in a smaller largest eigenvalue of (60), leading to a slower convergence rate and increased difficulty in training.

$$\boldsymbol{D}_\varepsilon = -\varepsilon^2 \Delta_{\mathrm{dis}} + \frac{1}{d}\mathrm{diag}\big(1, \ldots, 1\big) \tag{60}$$

### B.5. High Frequency Function Approximation

We aim to approximate the following function: $u = \sin(50\pi x), \quad x \in [0,1]$. The homotopy is defined as $H(u,\varepsilon) = u - \sin(\frac{1}{\varepsilon}\pi x)$, where $\varepsilon \in [\frac{1}{50}, \frac{1}{15}]$. Number of residual points $n_{\mathrm{res}} = 300$. In this example, we optimize using the Homotopy Loss. We set $\varepsilon_0 = \frac{1}{15}$ and $\varepsilon_n = \frac{1}{50}$, the list for $\{\varepsilon_i\}$ is $[\frac{1}{15}, \frac{1}{20}, \frac{1}{25}, \frac{1}{30}, \frac{1}{35}, \frac{1}{40}, \frac{1}{45}, \frac{1}{50}]$. From this example, we observe that the homotopy dynamics approach can also mitigate the slow training issue caused by the Frequency Principle (F-Principle) when neural networks approximate high-frequency functions.

As shown in Figure 14, due to the F-principle (Xu et al., 2024), training is particularly challenging when approximating high-frequency functions like $\sin(50\pi x)$. The loss decreases slowly, resulting in poor approximation performance. However, training based on homotopy dynamics significantly reduces the loss, leading to a better approximation of high-frequency functions. This demonstrates that homotopy dynamics-based training can effectively facilitate convergence when approximating high-frequency data. Additionally, we compare the loss for approximating functions with different frequencies $1/\varepsilon$ using both methods. The results, presented in Table 7, show that the homotopy dynamics training method consistently performs well for high-frequency functions.

*Table 7.* Comparison of the lowest loss achieved by the classical training and homotopy dynamics for different values of $\varepsilon$ in approximating $\sin\left(\frac{1}{\varepsilon}\pi x\right)$

|  | $\varepsilon = 1/15$ | $\varepsilon = 1/35$ | $\varepsilon = 1/50$ |
|---|---|---|---|
| Classical Loss | 4.91e-6 | 7.21e-2 | 3.29e-1 |
| Homotopy Loss $L_H$ | 1.73e-6 | 1.91e-6 | **2.82e-5** |

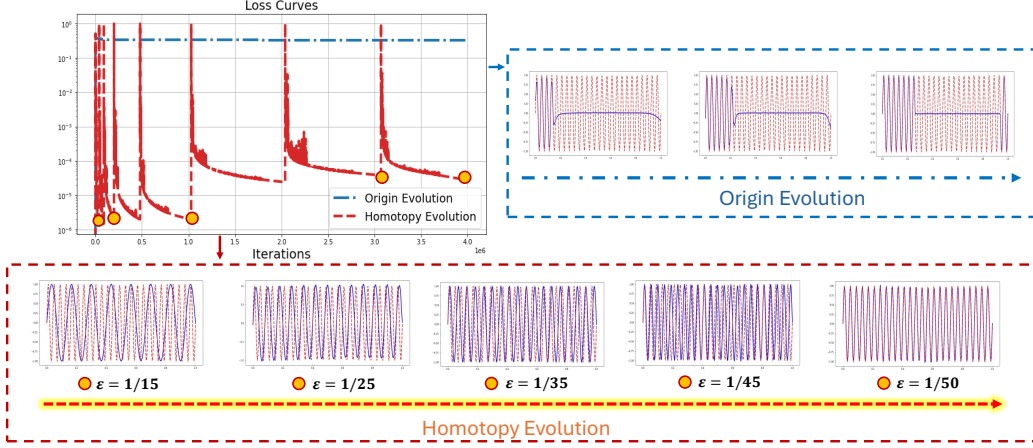

*Figure 14.* High-frequency function $\sin(50\pi x)$ approximation: Comparison of loss curves between original evolution and homotopy evolution. The comparison shows that homotopy evolution effectively reduces the loss, successfully approximating the high-frequency function, while the original evolution fails. The number of residual points is $n_{\text{res}} = 300$.

### B.6. Operator Learning 1D Burgers' Equation

In this example, we apply homotopy dynamics to operator learning. The neural network architecture follows the DeepONet structure:

$$\mathcal{G}_{\boldsymbol{\theta}}(v)(y) = \sum_{k=1}^{p}\sum_{i=1}^{n} a_i^k \sigma\left(\sum_{j=1}^{m}\xi_{ij}^k v\left(x_j\right)+c_i^k\right)\sigma\left(w_k \cdot y + b_k\right). \tag{61}$$

Here, $\sigma\left(w_k \cdot y + b_k\right)$ represents the trunk net, which takes the coordinates $y \in D'$ as input, and $\sigma\left(\sum_{j=1}^{m}\xi_{ij}^k u\left(x_j\right)+c_i^k\right)$ represents the branch net, which takes the discretion function $v$ as input. Rigorous error bounds for DeepONet are established in (Lanthaler et al., 2022; Liu et al., 2024a; Yang, 2024), so we omit them here. We can interpret the trunk net as the basis functions for solving PDEs. For this example, the input is $u_0$ and the output is $u_\infty$. We still train using the homotopy loss. It is important to emphasize that, unlike conventional operator learning, which typically follows a supervised learning strategy, our approach adopts an unsupervised learning paradigm. This makes the training process significantly more challenging. The initial condition $u_0(x)$ is generated from a Gaussian random field with a Riesz kernel, denoted by GRF $\sim \mathcal{N}\left(0, 49^2(-\Delta + 49I)^{-4}\right)$ and $\Delta$ and $I$ represent the Laplacian and the identity. We utilize a spatial resolution of 128 grids to represent both the input and output functions.

We want to find the steady state solution for this equation and $\varepsilon = 0.05$. The homotopy is:

$$H(u, s, \varepsilon) = (1-s)\left(\left(\frac{u^2}{2}\right)_x - \varepsilon(s)u_{xx} - \pi\sin(\pi x)\cos(\pi x)\right) + s(u - u_0), \tag{62}$$

where $s \in [0, 1]$. In particular, when $s = 1$, the initial condition $u_0$ automatically satisfies and when $s = 0$ becomes the steady state problem. And $\varepsilon(s)$ can be set to

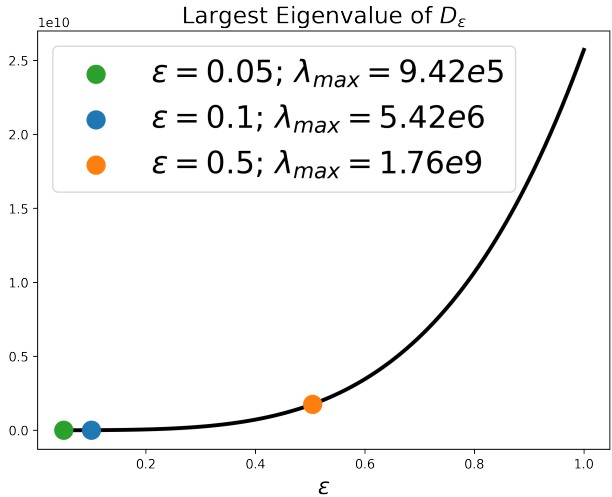

*Figure 15.* Largest eigenvalue of $\boldsymbol{D}_\varepsilon$ (64) for different $\varepsilon$. A smaller $\varepsilon$ results in a smaller largest eigenvalue of (64), leading to a slower convergence rate and increased difficulty in training.

*Table 8.* **Comparison of accuracy and efficiency between Finite Difference Method (FDM) and DeepONet (trained via Homotopy Dynamics).**

| $\varepsilon$ | $\Delta t$ | **Finite Difference Method (FDM)** | | | | **DeepONet (trained by Homotopy)** | | |
|---|---|---|---|---|---|---|---|---|
| | | L2RE | MSE $(x_s)$ | Comp. Time (s) | Loss $L_H$ | L2RE | MSE $(x_s)$ | Inference Time (s) |
| 0.5 | $5 \times 10^{-5}$ | 1.63e-12 | 7.35e-13 | 239.98 | 7.55e-7 | 1.50e-3 | 1.75e-8 | 0.2 |
| 0.1 | $1 \times 10^{-5}$ | 5.83e-4 | 1.57e-5 | 1239.77 | 3.40e-7 | 7.00e-4 | 9.14e-8 | 0.2 |
| 0.05 | $5 \times 10^{-6}$ | 1.01e-2 | 4.20e-3 | 2416.23 | 7.77e-7 | 2.52e-2 | 1.20e-3 | 0.2 |

$$\varepsilon(s) = \begin{cases} s, & s \in [0.05, 1], \\ 0.05 & s \in [0, 0.05]. \end{cases} \tag{63}$$

Here, $\varepsilon(s)$ varies with $s$ during the first half of the evolution. Once $\varepsilon(s)$ reaches 0.05, it is fixed at $\varepsilon(s) = 0.05$, and only $s$ continues to evolve toward 0.

**Largest eigenvalue of $\boldsymbol{D}_\varepsilon$.** As shown in Figure 15, a smaller $\varepsilon$ results in a smaller largest eigenvalue of (64), leading to a slower convergence rate and increased difficulty in training.

$$\boldsymbol{D}_\varepsilon = -\varepsilon^2 \Delta_{\text{dis}} + \text{diag}\left( u(\boldsymbol{x}_1)\frac{\mathrm{d}}{\mathrm{d}x_{\text{dis}}} + \frac{\mathrm{d}u(\boldsymbol{x}_1)}{\mathrm{d}x}, \ldots, u(\boldsymbol{x}_n)\frac{\mathrm{d}}{\mathrm{d}x_{\text{dis}}} + \frac{\mathrm{d}u(\boldsymbol{x}_n)}{\mathrm{d}x} \right). \tag{64}$$

**Compare with tradition method.**

The table reports both inference time and accuracy metrics across varying $\varepsilon$. While FDM achieves high accuracy, its computational cost increases significantly for small $\varepsilon$ due to CFL constraints. Moreover, its accuracy deteriorates under small $\varepsilon$, possibly due to resolution limitations. In contrast, our DeepONet model yields substantially faster inference with only moderate accuracy degradation, making it well-suited for many-query scenarios such as uncertainty quantification or real-time control.

