# OpenReview forum: "Learn Singularly Perturbed Solutions via Homotopy Dynamics"
_ICML.cc/2025/Conference — ICML 2025 poster_

### Official Review · Reviewer_fKP4 · 2025-02-17

**Overall Recommendation:** 2

**Summary:**

This paper introduces homotopy dynamics as a strategy to solve PDEs with sharp interfaces using PINNs. The key idea is to start training with a larger interface width parameter $\epsilon$ (corresponding to a smoother solution), then gradually decrease $\epsilon$ to the desired sharp-interface regime. This approach is particularly relevant for PDEs such as the Allen-Cahn equation, where $\epsilon$ controls the interface sharpness.

**Claims And Evidence:**

The claims are supported by the provided evidence, but the support is not particularly strong, see Experimental Designs and Analyses.

**Essential References Not Discussed:**

See Experimental Designs Or Analyses.

**Experimental Designs Or Analyses:**

The experimental setup is reasonable, but the baseline comparison is relatively weak.

- Original training are known to struggle with PDEs that have highly oscillatory or near-singular solutions. Prior works have proposed strategies to address this issue:
  - **Curriculum regularization**: Krishnapriyan et al. (2021) suggest first learning simpler problems before tackling harder ones, similar to the homotopy dynamics approach. They also propose learning early time steps first, which relates to the design of the homotopy in Examples 5.1 and 5.3 (where $s = 1$ corresponds to the initial condition and $s = 0$ to the full time problem).
  - **Neural Tangent Kernel Perspective**: Wang et al. (2021) analyze PINN training and show that PINNs are biased toward smooth solutions, which could be relevant to Theorem 4.1. The proposed remedy involves using random Fourier features.

The homotopy dynamics approach appears to provide a more principled way to implement some of these intuitions. However, a discussion on the similarities and differences between these methods, along with a performance comparison, would strengthen the paper.

1. Krishnapriyan, A.S., Gholami, A., Zhe, S., Kirby, R.M., Mahoney, M.W., 2021. Characterizing possible failure modes in physics-informed neural networks.
2. Wang, S., Wang, H., Perdikaris, P., 2021. On the eigenvector bias of Fourier feature networks: From regression to solving multi-scale PDEs with physics-informed neural networks.

In addition, there is also a difference between the experiments and the proposed method. When describing the proposed method, only $\epsilon$ changes. However, for the experiments, another parameter $s$ is introduced, which also modify the residual loss.

**Methods And Evaluation Criteria:**

The proposed method and evaluation criteria are reasonable for the problem at hand. But there are alternative methods that are not explored or discussed:

- The bigger questions is, why do we use PINN for this type of problems instead of well-established numerical methods?
- Regarding the the homotopy loss $L_{H_\epsilon}$  (line 181). What happens if we don't use $L_{H_\epsilon}$? That is, we only change $\epsilon$. This would be similar to the curriculum regularization (Krishnapriyan et al. 2021).
- Since $L_{H_\epsilon}$ is supposed to mimic the forward Euler step in strategy 1, what happens if we don't use the residual loss $L_H$ and only use $L_{H_\epsilon}$?
- What is the effect of $\alpha$?

**Other Comments Or Suggestions:**

- Algorithm 1, line 199, why do wee need $\Delta \epsilon_k$ in the inner loop of strategy 2
- For the numerical examples, which strategy is used?

**Other Strengths And Weaknesses:**

Strengths
- The homotopy dynamics method is a principled and effective way to address challenges in training PINNs for sharp-interface PDEs.
- The paper is well-written and easy to follow, making it accessible to readers with different levels of familiarity with PINNs and interface PDE
- Theorems provide insights into why homotopy dynamics improves training.
- The approach has the potential to benefit researchers working on PINN, particularly in problems involving sharp interface.

Weakness:
- The baseline is relatively week, see Experimental Designs Or Analyses.
- Alternative strategies are not fully explored, see Methods And Evaluation Criteria.

**Questions For Authors:**

See previous comments, Methods And Evaluation Criteria, Experiment Design and Analysis.

**Relation To Broader Scientific Literature:**

The proposed method improve PINN for a specific type of PDE problem.

**Theoretical Claims:**

A few clarifications would improve clarity
- Equation (13) - (18), L should be $L_H$ in equation (12)? That is, the loss under analysis is the residual loss without homotopy dynamics?
- Shouldn't $l$ depends on $\epsilon$? That is, $l = l_\epsilon$?
- In the proof Appendix A.2, line 720 and 736, What is $K_\epsilon$? Is it $K_\epsilon(\theta(0))$?
- How do we go from 736 to 740? Is it Weyl's inequality? Please clarify.
- From theorem 4.1 and the discussion afterward, if the difficulty only lies in the speed of convergence, will a larger learning rate help for original training?
- Theorem 4.3, what is $K_\epsilon$? Is it related to the $K_\epsilon$ in Theorem 4.1? If not, using a different notation might improve clarity.
- As $\epsilon\rightarrow 0$, the solution becomes more singular. Does $K_\epsilon$ remain bounded? If not, then K might not exist (line 305 column 2).
- What is $N$ in Theorem 4.3?
- Appendix A.3, it seems (45) - (48) are more like standard analysis on Euler's method? How is it related to Antonakopoulos
et al., 2022?
- Theorem 4.3 is only related to strategy 1, where Euler's method is used to evolve $\theta$, but not strategy 2. The author should explain the connection between the two strategies.

---

> ### Author Rebuttal · Authors · 2025-03-31
>
> Thank you for your thoughtful review and constructive suggestions. Below, we address the concerns you raised.
> - **Why do we use PINN for this type of problem instead of well-established numerical methods?**
>   Neural networks offer strong approximation capabilities [1] and help mitigate the curse of dimensionality [2], making them well-suited for solving PDEs. They have been widely applied in this context [3,4], particularly in operator learning, where they can significantly accelerate computation. Moreover, many AI for Science models are governed by similar equations, and neural networks enable seamless integration of experimental data into the modeling process.
>
> - **Regarding the homotopy loss $L_{H_\varepsilon}$ (line 181):**
>   Without $L_{H_\varepsilon}$, simply varying $\varepsilon$ leads to instability and sensitivity to $\Delta \varepsilon$. The homotopy loss enables a stable Euler-forward-like path consistent with the homotopy dynamics. We will include supporting experiments in the revised version.
>
> - **What happens if we use only $L_{H_\varepsilon}$ and omit the residual loss $L_H$?**
>   A solution satisfying $H(u, \varepsilon) = \text{Const} \neq 0$ still minimizes $L_{H_\varepsilon}$. Thus the residual loss is necessary to enforce the target PDE constraint.
>
> - **What is the effect of $\alpha$?**
>   It balances the three loss terms, playing a similar role as $\lambda$.
>
> - **Algorithm 1, line 199: Why do we need $\Delta \epsilon_k$ in Strategy 2?**
>   It ensures proper scaling of $L_{H_\varepsilon}$ and guides the homotopy step correctly.
>
> - **For the numerical examples, which strategy is used?**
>   Strategy 1 is used for 1D Allen–Cahn and high-frequency examples. Strategy 2 is used for 2D Allen–Cahn and Burgers, where solving the linear system is more difficult.
> ---
> **Theoretical Answer:**
>
> - **Items 1–3:** We agree. $L$ should be $L_H$, $l$ as $l_\varepsilon$, and $K_\varepsilon$ refers to $K_\varepsilon(\theta(0))$. These will be corrected.
>
> - **Item 4:** Based on Weyl’s inequality:
>
>   $\lambda_{\min}(A+B) \ge \lambda_{\min}(A) + \lambda_{\min}(B)$
>   Therefore,
>   $\lambda_{\min}(K_\varepsilon(\theta(t)))
>   \ge \lambda_{\min}(K_\varepsilon(\theta(t)) - K_\varepsilon(\theta(0))) + \lambda_{\min}(K_\varepsilon(\theta(0)))$
>
>   $\ge \lambda_{\min}(K_\varepsilon(\theta(0))) - \sigma_{\max}(K_\varepsilon(\theta(t)) - K_\varepsilon(\theta(0)))$
>
>   $\ge \lambda_{\min}(K_\varepsilon(\theta(0))) - ||K_\varepsilon(\theta(t)) - K_\varepsilon(\theta(0))||_F$
>
>   $\ge \frac{1}{2}\lambda_{\min}(K_\varepsilon(\theta(0)))$
> We will add this in the revised proof.
>
> - **Item 5:** A large learning rate won’t solve the slow convergence issue for small $\varepsilon$, and may lead to instability [3].
>
> - **Item 6:** Thank you—we will update the notation.
>
> - **Item 7:** For small $\varepsilon$, $K_\varepsilon$ may be large but remains finite. Using small $\Delta \varepsilon$ ensures stable error control.
>
> - **Items 8–9:** The index should be lowercase $n$, and the correct reference is [4].
>
> - **Item 10:** Without $L_H$ and $\lambda L_{bc}$, Strategy 2 becomes a way to solve Eq. (7). However, due to small singular values in $H_u \nabla_\theta u$, solving directly may be unstable. Thus, we adopt optimization instead. Strategy 2 shares the same dynamics as Strategy 1. Including $L_H$ ensures the solution satisfies the PDE. Even if $H(u, \varepsilon) = \text{Const} \neq 0$, it still minimizes $L_{H_\varepsilon}$, hence both terms are necessary.
>
> ---
>
> **Comments on the Weaknesses:**
> We thank the reviewer for the insightful comments. Compared with [5], our method introduces a homotopy loss that prevents convergence to bifurcation solutions. This key idea is not present in [5]. Also, the difficulty of training when $\varepsilon \to 0$ is not theoretically addressed in [6]; we believe ours is the first rigorous analysis. Strategy 2 is an alternative solver within the same framework, so our theoretical results are based on Strategy 1.
>
> **Experimental Comparison:**
> Based on Example 5.1, we added comparisons with other methods. In this example, the homotopy is defined as $H(u,s,\epsilon(s))$. Results in [Table 3 and Figures 1–3](https://drive.google.com/file/d/1fvQdXIZpm0c3fci_85Sb4CW-ZIjVHBlx/view) show that our homotopy-based method consistently achieves better accuracy.
>
> ---
> **References:**
>
> [1] [Yang et al., NeurIPS, 2023.](https://proceedings.neurips.cc/paper_files/paper/2023/hash/449a016a6ce6fba3fe50d05482abf836-Abstract-Conference.html)
> [2] [E et al., Constructive Approximation, 2022.](https://arxiv.org/abs/1906.08039)
> [3] [Sutskever et al., ICML, 2013.](https://proceedings.mlr.press/v28/sutskever13.html)
> [4] [Atkinson et al., Wiley, 2009.](https://homepage.math.uiowa.edu/~atkinson/papers/NAODE_Book.pdf)
> [5] [Krishnapriyan et al., NeurIPS, 2021.](https://arxiv.org/abs/2109.01050)
> [6] [Wang et al., Comput. Methods Appl. Mech. Eng., 2021.](https://www.sciencedirect.com/science/article/pii/S0045782521002759)

---

### Official Review · Reviewer_m8xv · 2025-03-15

**Overall Recommendation:** 3

**Summary:**

In this paper, the authors present a training based method based on homotopy dynamics for handling sharp interface problems. The authors provide a proof of the convergence of the Homotopy dynamics for stable training. The experiments results demonstrate that the proposed method can help capture the sharp interfaces as well as approximating high-frequency functions.

**Claims And Evidence:**

The authors claim that the proposed method can improve the training process for sharp interface problems, high frequency function approximation and operator learning. These claims are validated by three examples in the experiments part. The authors also provide a theoretical proof for the convergence of Homotopy Dynamics for stable training.

**Essential References Not Discussed:**

n/a

**Experimental Designs Or Analyses:**

The experiments demonstrate the effectiveness of the proposed method on sharp interface problem. Here the problems have more shaper interface with smaller parameter **epsilon**. The experiments show that the proposed homotopy loss keeps low while classical loss increases dramatically as the parameter value decreases.

**Methods And Evaluation Criteria:**

The proposed methods are evaluated on three examples: 2D Allen Cahn Equation, High Frequency Function Approximation and Burgers Equation, which looks sensible to me.

**Other Comments Or Suggestions:**

n/a

**Other Strengths And Weaknesses:**

The paper is well-written and easy to follow. The detailed background materials provided helps enhance the reader’s comprehension of the paper.

**Questions For Authors:**

- The examples shown in this paper are restricted in small scale 1D or 2D scenarios. Just wondering how would the method performed when applied to larger 3D cases?
- What are the training/inference time costs for the proposed method? How sensitive are the time costs to the number of trainable parameters?
- In this paper, the authors investigate the scenarios where a single parameter $\epsilon$ quantifies how singular the system is. Does the proof still holds if there are multiple parameters? How would the method extend to handle these multi parameter scenarios?

**Relation To Broader Scientific Literature:**

The proposed method provides a training strategy for learning based method such as PINN, Operator learning to learn sharp interface problems.  This paper also present some intuitions on the training difficulties caused by certain parameters in PDEs learning.

**Theoretical Claims:**

While I have thoroughly reviewed the methodology presented in the paper, I did not perform an exhaustive line-by-line verification of all mathematical derivations and proofs.

---

> ### Author Rebuttal · Authors · 2025-03-31
>
> Thank you for your valuable suggestions on our paper.
>
> First, We would like to emphasize the main focus and contribution of our work. This paper addresses the core challenge of training neural networks to solve PDEs, particularly those involving sharp interfaces, where specific parameters in the PDE induce near-singularities and hinder optimization. **From a theoretical perspective, we provide a novel analysis of how such parameters affect the convergence behavior during training.** To overcome these difficulties, we propose a **homotopy dynamics-based training strategy** and rigorously establish its convergence properties. On the experimental side, we demonstrate that our method not only performs effectively on the 2D Allen–Cahn equation, but also alleviates the spectral bias commonly seen in neural network training. Furthermore, we show that this homotopy-based approach generalizes well to the operator learning setting, highlighting its versatility and broad applicability.
>
> Below, we provide our responses to the questions and concerns you have raised.
>
> - We appreciate the reviewer’s suggestion regarding the extension of our method to larger-scale 3D cases. While the current examples in the paper are primarily 1D or 2D, it is important to emphasize that Example 5.3 is set in the operator learning framework, which is inherently more complex and challenging than standard PDE regression tasks.
>
>   In particular, unlike most existing operator learning methods that are trained in a supervised manner (i.e., with access to input-output solution pairs), our setting adopts a fully unsupervised training strategy based on homotopy dynamics, where the model learns the solution operator solely from the PDE structure. This significantly increases the difficulty of the learning problem.
>
>   Despite this challenge, our method achieves competitive and accurate results, highlighting its potential applicability not only to standard PINNs but also to more complex, unsupervised operator learning tasks.
>
>   We thank the reviewer again for the helpful suggestion, and in the revised version of the paper, we will incorporate higher-dimensional (e.g., 3D) examples to further demonstrate the effectiveness of our approach.
>
>
> - Regarding **training time**, we provide additional details here. All experiments were conducted on a single RTX 3070 Ti GPU. The corresponding computation times for training each epoch are summarized in the [Table 2](https://drive.google.com/file/d/1fvQdXIZpm0c3fci_85Sb4CW-ZIjVHBlx/view)
>     The detailed settings of each numerical experiment, including all parameter choices, are provided in Appendix B. We would like to emphasize that although our training time may appear relatively longer and the training procedure more involved, it enables us to achieve significantly higher accuracy. This level of precision cannot be attained by other methods, regardless of how long they are trained.
>
>   Regarding **inference time**, we take Example 5.3 as a representative case in the operator learning setting. Specifically, we compare the inference efficiency of our trained DeepONet model with that of the traditional finite difference method, by solving 1,000 instances of the PDE using both approaches. The results are summarized in the [Table 1](https://drive.google.com/file/d/1fvQdXIZpm0c3fci_85Sb4CW-ZIjVHBlx/view).
>
>   As with other neural network-based methods, increasing the number of network parameters generally leads to longer training times. Thank you for your helpful suggestion — we will include these details and clarifications in the revised version of the paper.
>
> - The question you raised is very interesting. We believe that our proposed homotopy dynamics-based approach can be extended to cases involving multiple parameters. At present, our initial idea is that in the multi-parameter setting, the homotopy dynamics may need to update the parameters sequentially or in a coordinated manner. We consider this a promising direction and plan to explore it further as part of our future work.
>
> Finally, we would like to thank you again for your insightful comments and questions.

---

> > ### Comment · Reviewer_m8xv · 2025-04-05
> >
> > Thanks for the authors' response. I will keep my score unchanged.

---

> > > ### Author Response · Authors · 2025-04-06
> > >
> > > Thank you again for your response and valuable suggestions. We will include a high-dimensional numerical experiment in the revised version. The results are shown below.
> > >
> > >   $$
> > >   -\Delta u = f_{\varepsilon},~~ \mathbf{x}\in \Omega, \quad
> > >   u = g,~~ \mathbf{x}\in \partial\Omega, \\
> > >   $$
> > >   where $\Omega=[-1, 1]^d$, $f_{\varepsilon}(x)=\frac{1}{\epsilon^2}\frac{1}{d}(\sin(\frac{1}{d}\sum \limits_{i=1}^{d}x_i)-2)$, which admit the exact solution $u(x)=(\frac{1}{d}\sum \limits_{i=1}^{d}x_i)^2+\sin(\frac{1}{\varepsilon}\frac{1}{d}\sum \limits_{i=1}^{d}x_i)$.  We consider $d=20$ and $\varepsilon = \frac{1}{35}$. Here, we employ a neural network with 5 layers and 128 neurons per layer. The training dataset consists of 10,000 interior points and 2,000 boundary points. The model is trained for $10^6$ epochs. The results are presented below.
> > >
> > >   | Method | Original PINN | Multiscale PINN [1] ($\sigma=35$) | Homotopy |
> > >   | :----: | :-----------: | :-------------------------------: | :------: |
> > >   |  L2RE  |    1.00e00    |              9.98e-1              | 5.84e-3  |
> > >
> > > The results indicate that our method performs well even for high-dimensional problems.
> > >
> > > **Reference**
> > >
> > > [1] [Wang et al., Comput. Methods Appl. Mech. Eng., 2021.](https://www.sciencedirect.com/science/article/pii/S0045782521002759)

---

### Official Review · Reviewer_Vjtr · 2025-03-22

**Overall Recommendation:** 2

**Summary:**

The authors look at the physics-informed neural network (PINNs) setting of solving a PDE via minimizing the PDE residual. They look at cases where there are “sharp” interfaces (introducing near singularities). They propose a method based on homotopy dynamics, which involves starting with an easier to learn problem and then moving towards a harder to learn problem (where “easy” and “hard” are characterized by the parameters in the PDE). They show this on the 1D Burgers equation and 1D and 2D Allen-Cahn equations.

**Claims And Evidence:**

The authors claim that this method makes it easier to get better error on PDE problems with near singular features. They show this on 1D Burgers, and 1D and 2D Allen-Cahn equations and that this type of training (easy to hard parameters) gets better error than directly using the PINNs approach on the “hard” problem right away.

**Essential References Not Discussed:**

Work that proposes a similar idea, as well as other works that attempt to deal with the same problems (such as Fourier features for high-frequency learning), are discussed above. There are many off-shoots and follow-ups of these works that are also relevant.

**Experimental Designs Or Analyses:**

The authors set up the three different PDE problems. They train a PINN with and without using the homotopy dynamics approach (directly solving the problem vs going from easy to hard parameters). See the above comment that there are now many methods to train PINNs better, many of which have addressed similar problems that the authors are looking at. At this point, it is needed to show proof-of-concept on much more difficult problems, such as those that many current PINNs methods struggle with.

**Methods And Evaluation Criteria:**

The method is based on the parameters of the PDE. They start with cases when the parameter in these different PDEs is higher, and the solution is easier to compute here. Then they train the model by going from this high parameter to the low parameter (and the original target problem).

However, these PDEs are still quite easy, and the PINNs literature has come a long way since. These systems were being studied years ago, with similar errors, and the field should be moving to harder problems at this point. Additionally, there are a number of other methods for improving PINNs training and prediction (including very similar ones) that are not compared to at all. Two examples:

This method looks very similar to the curriculum regularization approach in [1], where the authors started with an easy to learn PDE problem and then slowly trained the model to solve the harder problem. The authors try to demonstrate their approach on high frequency problems, but many approaches to address high frequency function approximation already exist, a simple one is to add Fourier features [2]. Additionally, if looking at the operator learning setting, how would it look to use something like Fourier Neural Operator [3] instead? These papers are many years old at this point, and the field has progressed a lot since: new approaches should be looking at much harder problems, comparing appropriately to prior approaches, and going beyond methods that have already been proposed.

The authors also mention resampling: there are also adaptive weighting methods that sample places where the PDE residual is high [4]. However, it seems like this approach is also not that computationally cheap since it requires training the network for longer by starting with the easier-to-learn parameter and then going to the harder one.

[1] Krishnapriyan et al. Characterizing possible failure modes of physics-informed neural networks. NeurIPS (2021)
[2] Wang, Wang, Perdikaris. On the eigenvector bias of Fourier feature networks. CMAME (2021)
[3] Li et al. Fourier Neural Operator. ICLR (2021)
[4] C. Wu, M. Zhu, Q. Tan, Y. Martha, L. Lu. A comprehensive study of non-adaptive and residual-based adaptive sampling for physics-informed neural networks. CMAME (2023).

**Other Comments Or Suggestions:**

See above for comments.

**Other Strengths And Weaknesses:**

See above for comments. The primary comments are that this work proposes something very similar to past work in the PINNs literature, doesn’t compare or contextualize against a vast literature of PINNs work that has been done to address many of the problems described here, and the experiment problems shown are relatively easy (compared to how far the field has come since).

**Questions For Authors:**

- How does this compare to adding Fourier features? How about the many other approaches that have been done for PINNs, such as the very similar curriculum regularization or adaptive sampling / adaptive weighting of the loss function?

- What is the computational cost of this method?

- These systems have been well-studied and are easy. Can these methods show proof-of-concept on much harder systems?

**Relation To Broader Scientific Literature:**

There is a lot of work on PINNs and better training methods for PINNs, as well as a wide range of work on using ML to solve PDEs. This work needs to be better contextualized against this broad landscape, and a lot of the progress that has been made in the field.

**Theoretical Claims:**

The authors present a theoretical analysis around the effectiveness of training PINNs via this homotopy dynamics approach. There are various places where a lot more steps would be helpful. There are also a lot of assumptions that are being made in these claims, such as only analyzing 2 layer neural networks, and looking at the width of the NN with ReLU activation functions. Does analysis based on ReLU activation functions actually apply here given that the neural network needs to be continuously differentiable to get derivatives and train with the PDE residual?

Also, how do you make the assumption in Eqn 31 in appendix?

---

> ### Author Rebuttal · Authors · 2025-03-31
>
> Thank you for your careful reading and valuable suggestions.
>
> First, we would like to emphasize the main motivation and contribution of our work.  **From a theoretical perspective, we analyze how such parameters affect training convergence speed**. To overcome these difficulties, we propose a **homotopy dynamics-based training strategy** with rigorous convergence analysis.
>
> Below, we address the questions and concerns you have raised.
>
> - **Comparision to Related Work**
>
>   We would like to emphasize that, to the best of our knowledge, **our work is the first to provide a theoretical justification that in sharp interface problems**, the parameter $\epsilon$ directly determines training difficulty—the smaller the $\epsilon$, the harder the optimization (see **Theorem 4.1**).
>
>   While our approach shares a high-level idea with curriculum-based methods (progressing from easy to hard tasks), it differs significantly in design. Unlike [1], which lacks a systematic mechanism, our homotopy dynamics defines a continuous path in the PDE parameter space with convergence guarantees. Moreover, we provide a dynamical update rule and a theoretically grounded strategy for choosing the homotopy step size $\Delta\epsilon$ (see **Theorem 4.3**), which is not addressed in [1].
>
>   Regarding Fourier feature methods [2], their success depends on prior knowledge and sensitive tuning of the feature scale $\sigma$. Our focus is on sharp interface problems, which differ from general multiscale settings. Example 5.2 shows that our method can generalize beyond sharp interfaces, demonstrating its versatility.
>
>   As for resampling-based methods [3,4], they often require large sample sizes and careful tuning, making them computationally expensive. In contrast, our approach achieves competitive accuracy with fewer collocation points and lower computational cost.
>
>   We further highlight these strengths in Example 5.1 (2D Allen–Cahn, $\epsilon = 0.05$). Unlike [1], which needs 50 time steps ($\Delta t = 0.1$), our homotopy strategy reaches the steady state in only 10 steps ($\Delta s = 0.1$) and uses just 2,500 collocation points. This demonstrates both the efficiency and effectiveness of our method.
>
>
>   Finally, we have added additional experimental comparisons, which further support the advantages of homotopy-based training and [Results](https://drive.google.com/file/d/1fvQdXIZpm0c3fci_85Sb4CW-ZIjVHBlx/view) show that our homotopy-based method consistently achieves better accuracy.
>
>  - **Theoretical Clarifications**
>
>     We appreciate the reviewer’s comments regarding theoretical assumptions and will incorporate further details in the revised version.
>
> 1. **Network Depth and Generality**: While we present the theory using two-layer networks for simplicity, our framework can be readily extended to deep architectures, building on standard results such as those in [2]. Our theoretical analysis focuses on how a small $\varepsilon$ induces optimization difficulties, regardless of the network depth, and the results remain valid. We use shallow networks solely to simplify the notation and enhance readability, thereby helping readers grasp our key points.
>
> 2. **Activation Functions**: Although we use ReLU in our theoretical analysis, the results hold for other smooth activations such as $\tanh$ and $\text{ReLU}^k$ (except for Lemma A.1, which has a known analog in [2]). In the revision, we will clarify that our results are **not restricted to ReLU**, and we will present a more general analysis accordingly.
>
> 3. **Clarification of Eq. (31)**: We note that Eq. (31) is **not an assumption**, but rather defines the **continuous kernel limit** as $m \to \infty$ (based on Law of Large Numbers). We will provide further explanation in the appendix to clarify this point.
>
> - **Other Questions**
>
> - Based on Example 5.1, we added comparisons with other methods. Results in [Table 3 and Figures 1–3](https://drive.google.com/file/d/1fvQdXIZpm0c3fci_85Sb4CW-ZIjVHBlx/view) show that our homotopy-based method consistently achieves better accuracy.
>
> - The training time and inference time for the numerical experiments can be found in  [Table 1 and 2](https://drive.google.com/file/d/1fvQdXIZpm0c3fci_85Sb4CW-ZIjVHBlx/view).
>
> - Our setting includes not only single PDEs, but also unsupervised operator learning, which is harder than the commonly studied supervised setup. Prior works [3,4] have explored this, but our method achieves higher accuracy. We believe our homotopy strategy can be extended to even more complex systems in future work.
>
>
> **References**
>
> [1] [Krishnapriyan et al., NeurIPS, 2021.](https://arxiv.org/abs/2109.01050)
> [2] [Wang et al., Comput. Methods Appl. Mech. Eng., 2021.](https://www.sciencedirect.com/science/article/pii/S0045782521002759) \
> [3] [Zhang et al., J. Comput. Phys., 2024.](https://doi.org/10.1016/j.jcp.2023.112638) \
> [4] [Li et al., Comput. Methods Appl. Mech. Eng., 2023.](https://doi.org/10.1016/j.jcp.2025.113843)

---

> > ### Comment · Reviewer_Vjtr · 2025-04-02
> >
> > Thank you for the response. I maintain concerns that the examples looked at here are too toy, as they have been well-studied for years now. Additionally, it would be useful to see more discussion on how the baseline comparisons were set up, and/or any code to compare these. For the Fourier features point, there is an experiment that explicitly relies on trying to capture high-frequency features, so Fourier features and other multi-scale approaches are a natural comparison.
> >
> > For the training time per epoch, the useful thing would be a total comparison of time trained, etc. against the speed of a numerical solver given the same accuracy. I think these examples are toy enough that a numerical solver will be faster.

---

> > > ### Author Response · Authors · 2025-04-06
> > >
> > > Thank you again for your response and valuable suggestions. In response to your concern, we would like to make the following clarifications.
> > >
> > > - **Comparison with tradition numerical method**
> > >
> > >   We conducted a detailed comparison between the finite difference method (FDM) and the DeepONet trained with our homotopy strategy by solving 1,000 instances of the Burgers' equation with varying initial conditions. We compared inference time, computational time, and relative $L^2$ error. As shown below, while FDM generally yields high accuracy, its computational cost rises sharply as $\varepsilon$ decreases due to CFL stability constraints. Moreover, its accuracy also degrades for small $\varepsilon$, likely due to resolution limitations.
> > >
> > >    |               |                  | **Finite Difference Method** (FDM) |             |                        |            | **DeepONet (trained by Homotopy)** |             |                   |
> > >   | :-----------: | :--------------: | ---------------------------------- | :---------: | ---------------------- | :--------: | :--------------------------------: | :---------: | ----------------- |
> > >   | $\varepsilon$ |    $\Delta t$    | L2RE                               | MSE distance($x_s$) | Computational Time (s) | Loss $L_H$ |               LE2RE                | MSE distance ($x_s$) | Inference Time(s) |
> > >   |      0.5      | $5\times10^{-5}$ | 1.63e-12                           |  7.35e-13   | 239.98                 |  7.55e-7   |              1.50e-3               |   1.75e-8   | 0.2               |
> > >   |      0.1      | $1\times10^{-5}$ | 5.83e-4                            |   1.57e-5   | 1239.77                |  3.40e-7   |              7.00e-4               |   9.14e-8   | 0.2               |
> > >   |     0.05      | $5\times10^{-6}$ | 1.01e-2                            |   4.20e-3   | 2416.23                |  7.77e-7   |              2.52e-2               |   1.2e-3    | 0.2               |
> > >
> > >
> > > - **High dimension case**
> > >
> > >   We will include a high-dimensional numerical experiment in the revised version. The results are shown below.
> > >
> > >   $$
> > >     -\Delta u = f_{\varepsilon},~~ \mathbf{x}\in \Omega \quad
> > >     u = g,~~ \mathbf{x}\in \partial\Omega,
> > >   $$
> > >     where $\Omega=[-1, 1]^d$, $f_{\varepsilon}(x)=\frac{1}{\epsilon^2}\frac{1}{d}(\sin(\frac{1}{d}\sum \limits_{i=1}^{d}x_i)-2)$, which admit the exact solution $u(x)=(\frac{1}{d}\sum \limits_{i=1}^{d}x_i)^2+\sin(\frac{1}{\varepsilon}\frac{1}{d}\sum \limits_{i=1}^{d}x_i)$.  We consider $d=20$ and $\varepsilon = \frac{1}{35}$. Here, we employ a neural network with 5 layers and 128 neurons per layer. The training dataset consists of 10,000 interior points and 2,000 boundary points. The model is trained for $10^6$ epochs. The results are presented below.
> > >
> > >   | Method | Original PINN | Multiscale PINN [1] ($\sigma=35$) | Homotopy |
> > >   | :----: | :-----------: | :-------------------------------: | :------: |
> > >   |  L2RE  |    1.00e00    |              9.98e-1              | 5.84e-3  |
> > >
> > >   The results indicate that our method performs well even for high-dimensional problems. For this high-dimensional problem, traditional numerical methods face significant challenges, making neural network-based approaches naturally advantageous. In our comparison, we observe that even the Fourier feature-based multiscale PINN [1] struggles to handle high-dimensional, high-frequency problems effectively. In contrast, our proposed homotopy-based training method achieves notably higher accuracy.
> > >
> > >
> > > - **Comparison with Multiscale PINN [1]**
> > >
> > > ​	Thank you for the suggestion. We compared with the Multiscale PINN in Example 5.2, which approximates a one-dimensional high-frequency function. Using the same network architecture, it achieves an MSE of $9.89 \times 10^{-8}$ at $\sigma = 30$, outperforming our method. This is likely due to its built-in basis functions $\sin(\sigma \pi x)$, which align well with targets like $\sin(50 \pi x)$. However, as shown in the high-dimensional Poisson example, its performance degrades significantly in more complex, high-dimensional settings.
> > >
> > > - **Setting for baseline model**
> > >
> > >   Due to space constraints, we omitted detailed baseline settings in the response. For clarity, the baseline models share the same network architecture, sample points, and training epochs as our homotopy-based method. Code and further implementation details will be included in the revised paper.
> > >
> > >   We have also provided supplementary information regarding the training time of our proposed method, as shown in the table below.
> > >
> > > |           Example            | 1D AC Equation | Example 5.1 | Example 5.2 | Example 5.3 |
> > > | :--------------------------: | :------------: | ----------- | :---------: | ----------- |
> > > | Training time for each epoch |     0.05s      | 0.09s       |    0.01s    | 0.4s        |
> > > |      Total epoch (step)      |     1.0e3      | 4.0e6       |    4.0e6    | 2.0e6       |
> > >
> > > [1] Wang et al., Comput. Methods Appl. Mech. Eng., 2021.

---

### Official Review · Reviewer_gjqL · 2025-03-25

**Overall Recommendation:** 3

**Summary:**

This paper proposes Homotopy Dynamics to train neural network for solving sharp interface problems. For sharp interface problems, the parameter $\epsilon$ in the PDE affects the singularity of the solution. As $\epsilon \to 0$, the PDE becomes increasingly singular and thus the solution is difficult to compute. The authors first train the
neural network on PDEs with a large $\epsilon$, and then adjust the neural network according to the evolution of the homotopy dynamics until $\epsilon$ decreases towards the target value. To validate the performance of the proposed methods, Numerical experiments on Allen-Cahn equation, high frequency function approximation, and Burgers equation are performed to validate the performance of the proposed method.

**Claims And Evidence:**

Most of the claims made in the submission are supported by clear and convincing evidence.

Question:

- In Theorem 4.1 the upper bound of $\lambda_{\min}(K_{\epsilon})$ is given. In line 301, 'Consequently, the training speed can reach $\exp(−Cn^3t$) based on Eq. (19), which is fast and implies that training is
easy.' This statement cannot be derived from Theorem 4.1 unless the bound is tight. More explanation is needed.

**Essential References Not Discussed:**

The paper adequately discusses the key related works necessary for understanding the context and its contributions.

**Experimental Designs Or Analyses:**

The experimental design appears sound and aligns well with the theoretical framework.

**Methods And Evaluation Criteria:**

The numerical experiments make sense for validating the performance of the proposed method.

**Other Comments Or Suggestions:**

Some typos:

- In line 94 right column, 'represent represent'

- In line 182 left column, the indices of $u$ are not consistent.

**Other Strengths And Weaknesses:**

Strengths:

- The proposed method improve the accuracy of the solution compared with other neural network-based methods.

Weaknesses:

- The motivation of using neural network to solve this problem is not convincing enough.

- In line 40. 'Leveraging neural network architectures to solve PDEs,... particularly in handling complex domains and incorporating empirical data'. Both these aspects were not emphasized throughout the paper.

**Questions For Authors:**

- As the neural network is designed and trained for a specific PDE, I think comparison with traditional methods should be included. For example, the computation cost (training time for using neural network versus the computation time using numerical solvers to solve the nonlinear equation); the accuracy of the solution, etc... to highlight the advantage of the proposed method.

- In example 5.3, why not directly train and solve for the steady state solution since it is independent of the initial condition.

**Relation To Broader Scientific Literature:**

The paper overcomes the optimization challenges lies in the training of neural network for solving PDEs; as introduced in Section 1. However, the motvation of using neural network to solve these PDEs are not adequately discussed. To my knowledge, the numerical solution of the example problems are well-studied using method other than neural network, to name just a few literature: some references in the paper including [Kreiss & Kreiss, 1986], [Hao & Yang, 2019] and

Kim, Yongho, Gilnam Ryu, and Yongho Choi. "Fast and accurate numerical solution of Allen–Cahn equation." Mathematical Problems in Engineering 2021, no. 1 (2021): 5263989.

Shen, Jie, and Xiaofeng Yang. "Numerical approximations of allen-cahn and cahn-hilliard equations." Discrete Contin. Dyn. Syst 28, no. 4 (2010): 1669-1691.

Jiwari, Ram. "A hybrid numerical scheme for the numerical solution of the Burgers’ equation." Computer Physics Communications 188 (2015): 59-67.

**Theoretical Claims:**

I did not carefully check the correctness of proofs for theoretical claims in Appendix A.

---

> ### Author Rebuttal · Authors · 2025-03-31
>
> Thank you for your careful reading and valuable suggestions. Our work addresses the core challenge of training neural networks to solve PDEs with sharp interfaces, where small parameters introduce near-singularities that hinder optimization. **From a theoretical perspective, we analyze how such parameters affect training convergence.** To overcome these difficulties, we propose a **homotopy dynamics-based training strategy** with rigorous convergence analysis. Experimentally, we show the method performs effectively on the 2D Allen–Cahn equation, mitigates spectral bias, and generalizes well to the operator learning setting.
>
> In response to the concerns you raised, we provide the following answers:
>
> - **Motivation for using neural networks.** Neural networks are widely used as PDE solvers due to their strong approximation power[1] and ability to mitigate the curse of dimensionality [2]. They also benefit from automatic differentiation for efficient derivative computation. However, training neural networks for complex PDEs remains difficult. Our work targets these optimization challenges with a homotopy-based solution.
>
>   For time-dependent PDEs like Allen–Cahn, computing the steady-state solution ($T \to \infty$) with traditional solvers requires very small $\Delta t$ as $\epsilon \to 0$, making them expensive. Neural networks, once trained, allow efficient inference—especially on GPUs. In Example 5.3, we add a comparison between our homotopy-trained DeepONet and classical finite difference methods across 1,000 equations (see [Table 1](https://drive.google.com/file/d/1fvQdXIZpm0c3fci_85Sb4CW-ZIjVHBlx/view?usp=sharing)), demonstrating significant computational speedups.
>
>   Neural network methods for these PDEs have gained attention recently[3,4,5]. Many physical models (e.g., phase-field dynamics) are governed by Allen–Cahn-type equations. Neural networks provide a path to integrate real experimental data into more accurate physical models.
> - **Clarification on Theorem 4.1.** In Theorem 4.1, we present two inequalities. The first inequality (Eq. (19)) characterizes the worst-case scenario, attaining equality when all components vanish except the one associated with the smallest eigenvalue. In practice, however, the decay of the other components is typically faster, rendering Eq. (19) nearly tight. An eigenvalue decomposition reveals that components along other eigen-directions diminish more rapidly, so that the smallest-eigenvalue component dominates after a short transient period.
>
>   For the second inequality (Eq.(20)), the Lidskii–Mirsky–Wielandt theorem provides the following full inequality:
>
>   $$
>   \lambda_{\text{min}}(D_\varepsilon^{T} D_\varepsilon)\lambda_{\text{min}}(S^{T} S) \le \lambda_{\text{min}}(K_\varepsilon) \le
>   \lambda_{\text{max}}(D_\varepsilon^{T} D_\varepsilon)\lambda_{\text{min}}(S^{T} S)
>   $$
>
>   Thus, when $\varepsilon$ is large, the training speed ranges from $\exp(-Ct/n)$ to $\exp(-Ctn^3)$. For small $\varepsilon$, it consistently decays as $\exp(-Ct/n)$.  The upper bound of the above inequality is attained in specific cases where a nonzero vector $x$ exists such that it is the eigenvector of the largest eigenvalue of $D_\varepsilon^\top D_\varepsilon$, and $D_\varepsilon x$ is the eigenvector of the smallest eigenvalue of $S^\top S$. This occurs under certain $S$ and $D_\varepsilon$ structures, depending on the PDE and sampling distribution. Hence, when $\varepsilon$ is large, training can be relatively easy in some cases; however, when $\varepsilon$ is small, training becomes universally difficult.
> - **Clarification on Example 5.3.** The objective in this operator learning task is to map the initial condition to its corresponding steady-state solution. In Example 5.3, we select a setup where the steady state is identical across initial conditions. This simplifies verification of whether the operator correctly maps diverse inputs to the same target. However, our homotopy-based training is not restricted to such cases and can be applied to settings where steady states differ. This example serves as a proof-of-concept showing the method's effectiveness even in an unsupervised operator learning setting, contrasting with the typical supervised approaches.
>
> Finally, we thank the reviewer for the constructive feedback and helpful suggestions, which have greatly improved the clarity and completeness of our paper.
>
> **Reference:**
> - [1][Yang et al., NeurIPS, 2023.](https://arxiv.org/abs/2305.08466)
> - [2][E et al., Constructive Approximation, 2022.](https://arxiv.org/abs/1906.08039)
> - [3][Wight et al., Commun. Comput. Phys., 2021.](https://arxiv.org/abs/2007.04542)
> - [4][Zhang et al., J. Comput. Phys., 2024.](https://doi.org/10.1016/j.jcp.2023.112638)
> - [5][Li et al., Comput. Methods Appl. Mech. Eng., 2023.](https://doi.org/10.1016/j.jcp.2025.113843)

---

> > ### Comment · Reviewer_gjqL · 2025-04-03
> >
> > Thank you for addressing my questions and concerns. I have increased my score accordingly.
> >
> > ------------------- Below are updated before April 7 --------------------------------
> >
> > Thank you for the response. I have some follow-up questions and comments:
> >
> > 1. In Table 1, what is the accuracy of the numerical solvers? I am asking because usually the numerical methods solves to a relative high accuracy. Probably it is not fair (and not necessary) to compare inference time against solver time.
> >
> > 2. In table 2, training time for each epoch is listed, how many epochs in total for training towards target epsilon for each example?
> >
> > 3. As the advantage of neural network approach is to overcome the curse of dimensionality, it would be helpful to see some examples (high dimension and large scale) that classical methods cannot handle.

---

> > > ### Author Response · Authors · 2025-04-06
> > >
> > > - We appreciate the reviewer’s concern regarding the fairness of comparing inference time with numerical solver time. To clarify, our intention in Table 1 is not to suggest that DeepONet can fully replace classical numerical solvers in all scenarios, but rather to demonstrate the potential efficiency gains in the **operator learning** setting when solving a large number of PDE instances.
> > >
> > >   In our revised Table 1 as shown below, we include both the inference time and corresponding accuracy metrics for the DeepONet model and the traditional finite difference method (FDM). As shown, although FDM typically achieves high accuracy, its computational cost increases significantly as $\varepsilon$ decreases, due to the stability constraints imposed by the CFL condition. At the same time, we observe that the accuracy of FDM also deteriorates under small $\varepsilon$, possibly due to resolution limitations.
> > >
> > >   In contrast, the DeepONet trained via our proposed homotopy dynamics strategy offers **substantially faster inference** across all tested settings, with **only moderate degradation in accuracy**. This efficiency–accuracy trade-off highlights the advantage of using DeepONet in contexts where many-query evaluations are required, such as uncertainty quantification or real-time control.
> > >
> > >
> > >
> > >   |               |                  | **Finite Difference Method** (FDM) |             |                        |            | **DeepONet (trained by Homotopy)** |             |                   |
> > >   | :-----------: | :--------------: | ---------------------------------- | :---------: | ---------------------- | :--------: | :--------------------------------: | :---------: | ----------------- |
> > >   | $\varepsilon$ |    $\Delta t$    | L2RE                               | MSE distance($x_s$) | Computational Time (s) | Loss $L_H$ |               LE2RE                | MSE distance ($x_s$) | Inference Time(s) |
> > >   |      0.5      | $5\times10^{-5}$ | 1.63e-12                           |  7.35e-13   | 239.98                 |  7.55e-7   |              1.50e-3               |   1.75e-8   | 0.2               |
> > >   |      0.1      | $1\times10^{-5}$ | 5.83e-4                            |   1.57e-5   | 1239.77                |  3.40e-7   |              7.00e-4               |   9.14e-8   | 0.2               |
> > >   |     0.05      | $5\times10^{-6}$ | 1.01e-2                            |   4.20e-3   | 2416.23                |  7.77e-7   |              2.52e-2               |   1.2e-3    | 0.2               |
> > >
> > >
> > >
> > > - Thank you for the suggestion. We have also included the total number of training epochs required. All training in our experiments is performed using full-batch training.
> > >
> > > |           Example            | 1D AC Equation | Example 5.1 | Example 5.2 | Example 5.3 |
> > > | :--------------------------: | :------------: | ----------- | :---------: | ----------- |
> > > | Training time for each epoch |     0.05s      | 0.09s       |    0.01s    | 0.4s        |
> > > |      Total epoch (step)      |     1.0e3      | 4.0e6       |    4.0e6    | 2.0e6       |
> > >
> > > - Thank you very much for your valuable suggestion. Indeed, we have conducted a high-dimensional numerical experiment, as shown below.
> > >   $$
> > >   -\Delta u = f_{\varepsilon},~~ \mathbf{x}\in \Omega,\\
> > >   \quad u = g,~~ \mathbf{x}\in \partial\Omega, \\
> > >   $$
> > >   where $\Omega=[-1, 1]^d$, $f_{\varepsilon}(x)=\frac{1}{d}(\sin(\frac{1}{\epsilon^2}\frac{1}{d}\sum \limits_{i=1}^{d}x_i)-2)$, which admit the exact solution $u(x)=(\frac{1}{d}\sum \limits_{i=1}^{d}x_i)^2+\sin(\frac{1}{\varepsilon}\frac{1}{d}\sum \limits_{i=1}^{d}x_i)$. We consider $d=20$ and $\varepsilon = \frac{1}{35}$. Here, we employ a neural network with 5 layers and 128 neurons per layer. The training dataset consists of 10,000 interior points and 2,000 boundary points. The model is trained for $10^6$ epochs. The results are presented below.
> > >
> > >   | Method | Original PINN | Multiscale PINN [1] ($\sigma=35$) | Homotopy |
> > >   | :----: | :-----------: | :-------------------------------: | :------: |
> > >   |  L2RE  |    1.00e00    |              9.98e-1              | 5.84e-3  |
> > > The results indicate that our method performs well even for high-dimensional problems.
> > >
> > > **Reference**
> > >
> > > [1] [Wang et al., Comput. Methods Appl. Mech. Eng., 2021.](https://www.sciencedirect.com/science/article/pii/S0045782521002759)

---

### Decision · Program_Chairs · 2025-05-01

**Decision:**

Accept (poster)

**Comment:**

This paper proposes an interesting approach that incorporates $\epsilon$, representing how sharp the phase transition in phase-field PDEs, into the training algorithm of a neural network approximator of the solution to phase-field problems. By the average score this is a typical borderline paper. The authors responded in the rebuttal with lots of extra experiments, however, I think this is an indication that this paper is not ready, or at least not ready for the targeted audiences. Here are a summary of questions or points I found in the reviews (including the comments in the AC-reviewer discussion) that would be helpful for a revision.

- How users should choose between the two strategies given in Algorithm 1? Is there any comparison?
- What is the strategy for choosing or designing the homotopy schedule? or any comparison or ablation on this matter.
- Some reviewers think that the experiments featured in this paper are "too easy", while I personally strongly disagree with this view, as in good science the experiments are ought to be illustrative of the theory. With this being said of my personal opinion, I respect the peer review process, and I would argue that adding baselines from previous PINN papers studying Allen-Cahn or Cahn-Hillard, such as [1] and [2], would make the case stronger.

Aside from reviewers' comments, I read the paper in details myself as well, here are my main concerns:
- Figure 1 definitely has some eye-catching drawing that is quite illustrative, however, as indicated by the figure, calling the SGD directly would result in "stuck in local minima", there is not a detailed ablation on this matter comparing the new training with a baseline training. In Figure 7, the caption reads "all final test solutions converge to the correct steady-state solution", yet does the vanilla SGD applying on the residual loss converge to an "incorrect" solution? This question remains unanswered throughout the paper. In Figure 2 there is a comparison of the loss curves of different epsilons, however, does a vanilla training strategy converge to an "incorrect" solution? or the large error is attributed simply to the solution being increasingly unsmooth?
- There is not an inner loop in the training algorithm, the $u_{\epsilon(s)} \to u_{\epsilon}$ is handled automatically by the homotopy dynamics. This is already vastly superior to most tailored training for PINN I read. However, the authors did not highlight this advantage over previous PINN training literature, for example [3].

Overall I think the idea is very good, and I have never seen anyone done this before (despite some of the reviewers' criticism). I believe this paper will be a very good contribution if the comments above can be added or addressed in a revision.


Minor typos:
- Page 2: "Homotopy dynamic"->"homotopy dynamics", and "the" should be capitalized after that.
- Page 6: "dynamic system"-> "dynamical system".
- Page 7: "a L2 error" -> "an L2 error"

---
[1]: C. Wight, J. Zhao. Solving Allen-Cahn and Cahn-Hilliard Equations using the Adaptive Physics Informed Neural Networks.

[2]: R. Mattey, S. Ghosh. A novel sequential method to train physics informed neural networks for Allen Cahn and Cahn Hilliard equations. CMAME (2022)

[3]: S. Wang et al. Understanding and mitigating gradient pathologies in physics-informed neural networks. SIAM Journal on Scientific Computing (2021)